# NeuralQP: A General Hypergraph-based Optimization Framework for Large-scale Quadratically Constrained Quadratic Programs

## Abstract

Machine Learning (ML)-based optimization frameworks have drawn increasing attention for their remarkable ability to accelerate the optimization procedure of large-scale Quadratically Constrained Quadratic Programs (QCQPs) by learning the shared problem structures, resulting in improved performance compared to classical solvers. However, current ML-based frameworks often struggle with strong problem assumptions and high dependence on large-scale solvers. This paper presents a promising and general hypergraph-based optimization framework for large-scale QCQPs, called NeuralQP. The proposed method comprises two key components: Hypergraph-based Neural Prediction, which generates the embedding of an arbitrary QCQP and obtains the predicted solution without any problem assumption; Iterative Neighborhood Optimization, which uses a McCormick relaxation-based repair strategy to quickly identify illegal variables in the predicted solution and iteratively improves the current solution using only a small-scale solver. Experiments on three classic benchmarks demonstrate that NeuralQP converges significantly faster than the state-of-the-art solves (e.g. Gurobi), further validating the efficiency of the ML-based framework for QCQPs.

## 1 Introduction

Quadratically Constrained Quadratic Programs (QCQPs) are mathematical optimization problems characterized by the presence of quadratic terms, finding extensive applications across diverse domains such as finance (Gondzio & Grothey, 2007), robotic control (Galloway et al., 2015), and power grid operations (Zhang et al., 2013). However, solving QCQPs is exceptionally challenging, especially for large-scale QCQPs, due to their discrete (Balas, 1969) and nonconvex nature (Elloumi & Lambert, 2019). With the advancement of machine learning (ML), the ML-based QCQP optimization framework has emerged as a promising research direction as it can effectively leverage the structural commonality among similar QCQPs to accelerate the solving process.

Current widely adopted ML-based QCQP optimization frameworks can be categorized into two types: *Solver-based Learning* and *Model-based Learning*. *Solver-based Learning* methods learn to tune the parameters or status of the solver to accelerate the solution process. As a representative work, RLQP Ichnowski et al. (2021) learned a policy to tune parameters of OSQP solver (Stellato et al., 2017) with reinforcement learning (RL) to accelerate convergence. Bonami et al. (2018) learned a classifier that predicts a suitable solution strategy on whether or not to linearize the problem for the CPLEX solver. Ghaddar et al. (2022) and Kannan et al. (2023) both learned branching rules on selected problem features to guide the solver. Although solver-based methods have demonstrated strong performance on numerous real-world QCQPs, their effectiveness heavily relies on large-scale solvers and is constrained by the solver's solving capacity, resulting in scalability challenges.

*Model-based learning* methods employ a neural network model to learn the parameters of the QCQP models, aiming to translate the optimization problem into a multiclass classification problem, of which the results can accelerate the solution process of the original optimization problem. Bertsimas & Stellato (2020; 2021) learned a multi-class classifier for both solution strategies and integer variable values, proposing an online QCQP optimization framework that consists of a feedforward neural network evaluation and a linear system solution. However, these methods make strong as-

sumptions about the model parameters. They assume that multiple problem instances are generated by a single model and that the model parameters are shared across different problem instances, which limits their applicability in real-world problem-solving scenarios.

To address the above limitations, this paper proposes NeuralQP, a general hypergraph-based optimization framework for large-scale QCQPs, which can be divided into two stages. In *Hypergraph-based Neural Prediction*, a representation of QCQPs based on a variable relational hypergraph with initial vertex embeddings is created as a lossless representation of QCQPs. Then UniEGNN, an improved hypergraph convolution strategy, takes both hyperedge features and vertex features as inputs, leverages a vertex-hyperedge-vertex convolution strategy, and finally obtains neural embeddings of the variables in QCQPs. In *Iterative Neighborhood Search*, to acquire feasible solutions after multiple neighborhood solutions are merged (which is termed crossover), a new repair strategy based on the McCormick relaxation is proposed to quickly identify violated constraints. Then, improperly fixed variables are reintroduced into the neighborhood, which realizes a progressive updating of the neighborhood radius and effective correction of infeasible solutions via a small-scale optimizer.

To validate the effectiveness of NeuralQP, experiments are conducted on three benchmark QCQPs, and the results show that NeuralQP can achieve better results than the state-of-the-art solver Gurobi and SCIP in a fixed wall-clock time using only a small-scale solver with 30% of the original problem size. Further experiments indicate that NeuralQP can achieve the same solution quality in less than one-fifth of the solving time of Gurobi and SCIP in large-scale QCQPs, which verifies the efficiency of the framework in solving QCQPs. Our contributions are concluded as follows:

1. We propose NeuralQP, the first general optimization framework for large-scale QCQPs without any problem assumption by means of a small-scale solver, shedding light on solving general nonlinear programming using an ML-based framework.

2. In *Hypergraph-based Neural Prediction*, a new hypergraph-based representation is proposed as a complete representation of QCQPs and an enhanced hypergraph convolution strategy is employed to fully utilize hyperedge features.

3. In *Iterative Neighborhood Search*, a new repair strategy based on the McCormick relaxation is proposed for neighborhood search and crossover with small-scale optimizers.

4. Experiments show that NeuralQP can accelerate the convergence speed with a small-scale optimizer, and can significantly increase the solving size of QCQPs compared with existing algorithms, indicating its robust capability in addressing large-scale QCQPs.

## 2 PRELIMINARIES

### 2.1 QUADRATICALLY-CONSTRAINED QUADRATIC PROGRAM

A Quadratically-Constrained Quadratic Program (QCQP) is an optimization problem that involves minimizing (or maximizing) a quadratic objective function subject to quadratic constraints (Elloumi & Lambert, 2019). Formally, a QCQP is defined as follows:

$$\begin{aligned}
\min or \max f(\boldsymbol{x}) &= \boldsymbol{x}^{\mathrm{T}} \boldsymbol{Q}^0 \boldsymbol{x} + \left(\boldsymbol{r}^0\right)^{\mathrm{T}} \boldsymbol{x}, \\
\text{s.t.} \quad & \boldsymbol{x}^{\mathrm{T}} \boldsymbol{Q}^i \boldsymbol{x} + \left(\boldsymbol{r}^i\right)^{\mathrm{T}} \boldsymbol{x} \le b_i, \quad \forall i \in \mathcal{M}, \\
& l_i \le x_i \le u_i, \quad \forall i \in \mathcal{N}, \\
& x_i \in \mathbb{Z}, \quad \forall i \in \mathcal{I}.
\end{aligned} \tag{1}$$

In the above equation, $\mathcal{M}$, $\mathcal{N}$ and $\mathcal{I}$ are the index sets of constraints, variables and integer variables respectively. Let $n = |\mathcal{N}|$ and $\boldsymbol{x} = (x_1, x_2, \ldots, x_n) \in \mathbb{R}^n$ denotes the vector of $n$ variables, with $l_i$ and $u_i$ being the lower and upper bounds of $x_i$. For each $k \in \{0\} \cup \mathcal{M}$, $\boldsymbol{Q}^k \in \mathbb{R}^{n \times n}$, representing the coefficients of quadratic terms, is symmetric but not necessarily positive (semi-)definite. $\boldsymbol{r}^k$ is the coefficient vector of linear terms and $b_k$ denotes the right-hand side of the $k$-th constraint.

### 2.2 GRAPH REPRESENTATIONS FOR MILPS

Graph representations for (mixed-integer) linear program denoted as (MI)LP have been proposed to transform the MILP into a suitable input format for graph neural networks, accompanied by

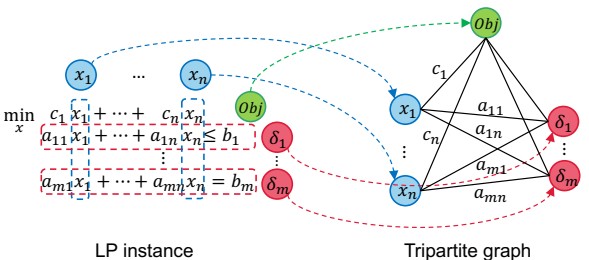 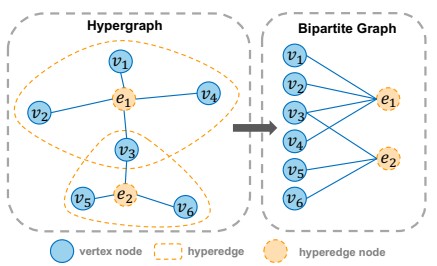

Figure 1: Tripartite representation. The green, blue, and red nodes represent the objective, variables, and constraints, and the edge features are associated with coefficients in the original problem.

Figure 2: Star expansion. A hypergraph is transformed into a bipartite graph, with nodes on the left and hyperedges on the right.

encoding strategies that further enhance the expressive power of the corresponding graph. This section introduces the specifics of *tripartite graph representation* and *random feature strategy*.

Gasse et al. (2019) first proposed a *bipartite graph representation* for MILPs that preserves the entire information of constraints, decision variables, and their relationships without loss. Ding et al. (2019) further proposed a *tripartite graph representation* for MILPs, which simplifies the feature representation of the graph by adding the objective node, ensuring that all coefficients appear only on edge weights. Figure 1 gives an example of the tripartite graph representation. In Figure 1, the left set of $n$ variable nodes represent decision variables with variable types and bounds encoded into node features; the $m$ constraint nodes on the right symbolize linear constraints with constraint senses and $b_i$ values encoded into node features; the objective node on the top represent the objective function with objective sense encoded as node features. The connecting edge $(i, j)$, signifies the presence of the $i$-th decision variable in the $j$-th constraint, with edge weight $a_{ij}$ representing the coefficient. The upper node signifies the objective, and the edge weight $c_i$ between the $i$-th variable node and the objective represents the objective coefficient.

However, there are MILP instances, named *foldable* MILPs (Chen et al., 2023) which have distinct optimal solutions yet the corresponding graph cannot be distinguished by the Weisfeiler-Lehman test (Weisfeiler & Leman, 1968). On these foldable MILPs, the performance of the graph neural networks could significantly deteriorate. To resolve such inability, Chen et al. (2023) proposed a *random feature strategy*, i.e., appending an extra dimension of random number to the node features, which further enhances the power of graph neural networks.

## 2.3 MCCORMICK RELAXATION

The McCormick relaxation (McCormick, 1976) is widely employed in nonlinear programming, aiming at bounding nonconvex terms by linear counterparts. Given variables $x$ and $y$ with bounds $L_x \leq x \leq U_x$ and $L_y \leq y \leq U_y$, the nonconvex term $\phi_{xy} := xy$ can be approximated by:

$$\phi_{xy} \leq \min\{L_y x + U_x y - L_y U_x, \ L_x y + U_y x - L_x U_y\},$$
$$\phi_{xy} \geq \max\{L_y x + L_x y - L_x L_y, \ U_y x + U_x y - U_x U_y\}, \quad (2)$$

which transforms the originally nonconvex problem into a more manageable linear format. Equation 2 can be derived by considering the expansions of the following four inequalities:

$$(U_x - x)(U_y - y) \geq 0, \quad (U_x - x)(y - L_y) \geq 0,$$
$$(x - L_x)(U_y - y) \geq 0, \quad (x - L_x)(y - L_y) \geq 0. \quad (3)$$

This method is particularly pertinent when dealing with nonconvex quadratic terms, for which obtaining feasible solutions is notably challenging due to the inherent nonlinearity and nonconvexity.

## 3 VARIABLE RELATIONAL HYPERGRAPH

The graph representation in Sec. 2.2 is limited to MILPs since it cannot represent the nonlinear terms. To address such limitation, we propose the *variable relational hypergraph* in this section to

model quadratic terms in QCQPs. To provide a rigorous definition, we first present the definition of hypergraphs (Sec. 3.1 & 3.2) and then formally define variable relational hypergraph (Sec. 3.3).

## 3.1 HYPERGRAPH BASICS

A hypergraph $\mathcal{H} = (\mathcal{V}, \mathcal{E})$ is defined by a set of vertices $\mathcal{V} = \{1, 2, \cdots, n\}$ and a set of hyperedges $\mathcal{E} = \{e_j\}_1^m$, where each hyperedge $e \in \mathcal{E}$ is a non-empty subset of $\mathcal{V}$ (Bretto, 2013). The incidence matrix $\boldsymbol{H} \in \{0, 1\}^{|\mathcal{V}| \times |\mathcal{E}|}$ is characterized by $\boldsymbol{H}(v, e) = 1$ if $v \in e$ else 0. A hypergraph is termed *k-uniform* (Rödl & Skokan, 2004) if every hyperedge contains exactly $k$ vertices. A distinctive class within hypergraph theory is the *bipartite* hypergraph (Annamalai, 2016). If $\mathcal{V}$ can be partitioned into two disjoint sets $\mathcal{V}_1$ and $\mathcal{V}_2$, a bipartite hypergraph is defined as $\mathcal{H} = (\mathcal{V}_1, \mathcal{V}_2, \mathcal{E})$ so that for each hyperedge $e \in E$, $|e \cap \mathcal{V}_1| = 1$ and $e \cap \mathcal{V}_2 \neq \emptyset$.

Hypergraphs can be converted into graphs via expansion techniques (Dai & Gao, 2023). We introduce the *star expansion* technique, which converts a hypergraph into a bipartite graph. Specifically, given a hypergraph $\mathcal{H} = (\mathcal{V}, \mathcal{E})$, each hyperedge $e \in \mathcal{E}$ is transformed into a new node in a bipartite graph. The original vertices from $\mathcal{V}$ are retained on one side of the bipartite graph, while each hyperedge is represented as a node on the opposite side, connected to its constituent vertices in $\mathcal{V}$ by edges, thus forming a star. Such transformation enables the representation of the high-order relationships in the hypergraph within the simpler structure of a bipartite graph. Figure 2 provides a bipartite graph representation obtained through the star expansion of a hypergraph.

## 3.2 HYEPRGRAPH MESSAGE PROPAGATION

To formalize the message propagation on hypergraphs, the Inter-Neighbor Relation (Dai & Gao, 2023) has been defined, which is essential in hypergraph learning models.

**Definition 1** *Inter-Neighbor Relation. The Inter-Neighbor Relation* $\mathrm{N} \subset \mathcal{V} \times \mathcal{E}$ *on a hypergraph* $\mathcal{H} = (\mathcal{V}, \mathcal{E})$ *with incidence matrix* $\boldsymbol{H}$ *is defined as:* $\mathrm{N} = \{(v, e) | \boldsymbol{H}(v, e) = 1, v \in \mathcal{V}, e \in \mathcal{E}\}$. *The hyperedge neighborhood* $\mathcal{N}_e(v)$ *of vertex* $v$ *and the vertex neighborhood* $\mathcal{N}_v(e)$ *of hyperedge* $e$ *are defined based on the Inter-Neighbor Relation.*

**Definition 2** *Hyperedge Neighborhood. The hyperedge neighborhood of vertex* $v \in \mathcal{V}$ *is defined as:* $\mathcal{N}_e(v) = \{e | v \mathrm{N} e, e \in \mathcal{E}\}$, *for each* $v \in \mathcal{V}$.

**Definition 3** *Vertex Neighborhood. The vertex neighborhood of hyperedge* $e \in \mathcal{V}$ *is defined as:* $\mathcal{N}_v(e) = \{v | v \mathrm{N} e, v \in \mathcal{V}\}$, *for each* $e \in \mathcal{E}$.

Using hypergraph Inter-Neighbor Relation, the general hypergraph convolution (also known as spatial convolution) (Gao et al., 2023) follows a vertex-hyperedge-vertex message propagation pattern. As the first representative framework of hypergraph convolution, *UniGNN* (Huang & Yang, 2021) utilizes a convolution method delineated in Equation 4:

$$\text{(UniGNN)} \begin{cases} h_e = \phi_1 \left( \{h_j\}_{j \in \mathcal{N}_v(e)} \right) \\ \tilde{h}_v = \phi_2 \left( h_v, \{h_i\}_{i \in \mathcal{N}_e(v)} \right) \end{cases}, \tag{4}$$

where $h_e$ symbolizes the hyperedge features aggregated from the vertices residing in hyperedge $e$; $h_v$ and $\tilde{h}_v$ represent the vertex features before and after the convolution respectively; the functions $\phi_1$ and $\phi_2$ are permutation-invariant functions. This spatial convolution strategy initiates by aggregating messages from the incident vertices of a hyperedge and then relays the aggregated message back to vertices, thus fulfilling a round of message propagation.

## 3.3 THE PROPOSED VARIABLE RELATIONAL HYPERGRAPH

Based on the definition of hypergraph, we propose the *variable relational hypergraph* as a lossless representation of QCQPs compatible with MILPs. Consider a QCQP problem defined in Equation 1, formal definitions are given below to describe the construction of such a hypergraph.

**Definition 4** *Extended variable vertex set.* *The variable vertex set $\mathcal{V}_x$ is defined as $\mathcal{V}_x = \{v_i | i \in \mathcal{N}\}$ and the extended variable vertex set $\bar{\mathcal{V}}_x$ is defined as $\bar{\mathcal{V}}_x = \mathcal{V}_x \cup \{v_0, v^2\}$, which is the variable vertex set with additional two vertices representing degree zero and degree two.*

**Definition 5** *Constraint vertex set and objective vertex set.* *The constraint vertex set $\mathcal{V}_c$ is defined as $\mathcal{V}_c = \{c_i | i \in \mathcal{M}\}$ and the objective vertex set $\mathcal{V}_o$ is defined as $\mathcal{V}_o = \{o\}$.*

**Definition 6** *V-O relational hyperedge.* *The set of V-O relational hyperedges is defined as $\mathcal{E}_o = \{\{v_i, v_0, o\} |$ if term $x_i$ is in the objective$\} \cup \{\{v_i, v^2, o\} |$ if term $x_i^2$ is in the objective$\} \cup \{\{v_i, v_j, o\} |$ if term $x_i x_j$ is in the objective$\}$.*

**Definition 7** *V-C relational hyperedge.* *The set of V-C relational hyperedges is defined as $\mathcal{E}_c = \{\{v_i, v_0, c_k\} |$ if term $x_i$ is in constraint $k\} \cup \{\{v_i, v^2, c_k\} |$ if term $x_i^2$ is in constraint $k\} \cup \{\{v_i, v_j, c_k\} |$ if term $x_i x_j$ is in constraint $k\}$.*

**Definition 8** *Variable relational hypergraph.* *A Variable Relational Hypergraph is defined as $\mathcal{H} = (\mathcal{V}_x, \mathcal{V}_o, \mathcal{V}_c, \mathcal{E}_o, \mathcal{E}_c)$ where $\mathcal{V}_x, \mathcal{V}_o, \mathcal{V}_c$ are sets of nodes representing variables, objective, and constraints respectively. $\mathcal{E}_o$ and $\mathcal{E}_c$ are sets of V-O relational and V-C relational hyperedges.*

From the above definitions, a variable relational hypergraph $\mathcal{H} = (\mathcal{V}_x, \mathcal{V}_o, \mathcal{V}_c, \mathcal{E}_o, \mathcal{E}_c)$ generated by MILPs or QCQPs is 3-uniform by the definition of relational hyperedges. Furthermore, let $\mathcal{V}_1 = \mathcal{V}_x, \mathcal{V}_2 = \mathcal{V}_c \cup \mathcal{V}_o$ and $\mathcal{E} = \mathcal{E}_c \cup \mathcal{E}_o$, then $\mathcal{H} = (\mathcal{V}_1, \mathcal{V}_2, \mathcal{E})$ is also a bipartite hypergraph.

## 4 THE GENERAL HYPERGRAPH-BASED OPTIMIZATION FRAMEWORK FOR LARGE-SCALE QCQPS

On the strength of the variable relational hypergraph, this section presents our proposed general hypergraph-based optimization framework for large-scale QCQPs, named NeuralQP. The framework of NeuralQP (illustrated in Figure 3) consists of *neural prediction* (Sec. 4.1) which predicts a high-quality initial solution and *iterative neighborhood optimization* (Sec. 4.2) which optimizes the incumbent solution by neighborhood search and crossover.

### 4.1 NEURAL PREDICTION

During this stage, QCQPs are initially converted into *hypergraph-based representation* (Sec. 4.1.1). During the training process, our proposed *UniEGNN* (Sec. 4.1.2) is applied on multiple hypergraphs to learn the typical structures of these problems. The outputs are interpreted as a probability of the optimal solution values similar to Neural Diving (Nair et al., 2021), which are adopted as a heuristic for *iterative neighborhood search* in Sec. 4.2. Details are presented in Appendix E.1.

#### 4.1.1 HYPERGRAPH-BASED REPRESENTATION

Based on Sec. 3, each QCQP instance is transformed into a variable relational hypergraph. Then, coefficients of the terms are further encoded as features of the corresponding hyperedges; variable, constraint and objective vertex features are generated in the same way introduced in Sec. 2.2.

For a concrete example shown in Figure 4, the QCQP instance is first converted to a variable relational hypergraph with initial embeddings, where the $q_{11}^0 x_1^2$ term in the objective is represented as a hyperedge covering vertices $x_1$, $x^2$ and $obj$ with feature $q_{11}^0$; the term $q_{1n}^1 x_1 x_n$ in constraint $\delta_1$ is represented as a hyperedge covering vertices $x_1$, 1 and $\delta_1$ with $q_{1n}^1$ as hyperedge feature; and the $r_n^m x_n$ term in constraint $\delta_m$ is represented as a hyperedge with feature $r_n^m$ covering vertices $x_1$, $x_2$ and $\delta_m$. The initial vertex and hyperedge embeddings are generated in a similar way as the tripartite graph representation with the random feature strategy in Sec.2.2.

#### 4.1.2 UNiEGNN

With the hypergraph representation, our proposed *UniEGNN* further employs the features of both vertices and hyperedges as the neural network inputs, converts the hypergraph into a bipartite graph,

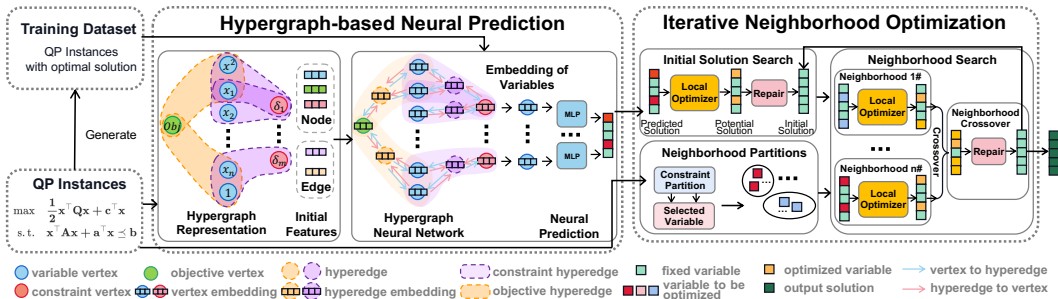

Figure 3: An overview of NeuralQP framework. The black line indicates the problem instances and their solutions. In the stage of hypergraph-based neural prediction, the QCQP is first encoded into a variable relational hypergraph with an initial node and hyperedge embeddings generated from the original problem. Then UniEGNN is applied to generate neural embeddings for each variable, utilizing both node and hyperedge features by converting the incumbent hypergraph into a bipartite graph. Then a multilayer perceptron layer predicts the optimal solution based on the neural embeddings. In the stage of iterative neighborhood optimization, the predicted solutions are first relaxed and repaired to get an initial feasible solution, after which neighborhood optimization with a small-scale optimizer is iteratively used. The iteration of neighborhood optimization consists of adaptive neighborhood partition, parallel neighborhood search, and McCormick relaxation-based neighborhood repair. The neighborhood search solution is again used as an initial feasible solution if the time limit is not reached; otherwise, the incumbent solution is output as the optimization result.

and updates vertex encodings via *spatial convolution* (Sec. 3.2). Firstly, different dimensional features of vertices and hyperedges are mapped into the same high-dimensional hidden space through a Multi-Layer Perceptron (MLP) layer. Then, vertices and hyperedges are reformulated as a bipartite graph via star expansion (Sec. 3) so that hyperedge features can be fully utilized in the spatial convolution. The bipartite graph contains two sets of nodes, $V$ nodes and $E$ nodes, which represent the original nodes and hyperedges. The new edge features correspond with the original hyperedge features, indicating that all edges connected with one $E$ node share the same feature. Details are illustrated in Figure 4. Finally, given the bipartite graph reformulation, a half-convolution strategy is employed as is detailed in Equation 5:

$$(\text{UniEGNN}) \begin{cases} h_e^{(t)} = \phi_e\left(h_e^{(t-1)}, \psi_e\left(\{h_\alpha^{(t-1)}\}_{\alpha \in \mathcal{N}_v(e)}\right)\right) \\ h_v^{(t)} = \phi_v\left(h_v^{(t-1)}, \psi_v\left(\{h_\beta^{(t)}\}_{\beta \in \mathcal{N}_e(v)}\right)\right) \end{cases}, \tag{5}$$

where $h_e^{(t)}$ and $h_v^{(t)}$ are hyperedge features and vertec features after the $t$-th convolution, respectively; $\phi_e$ and $\phi_v$ are two permutation-invariant functions implemented by MLP; $\phi_e$ and $\psi_v$ are aggregation functions for hyperedges and vertices, which are SUM and MEAN in NeuralQP respectively; $\mathcal{N}_v(e)$ and $\mathcal{N}_e(v)$ are the neighborhood of hyperedge $e$ and vertex $v$ defined in Sec. 3.2. Figure 5 illustrates the message propagation in the half-convolution process, where features of both $E$ nodes and $V$ nodes are updated. The red arrow indicates the flow to the specific node.

## 4.2 ITERATIVE NEIGHBORHOOD OPTIMIZATION

In this stage, a variable proportion $\alpha_{ub} \in (0,1)$ is defined so that a large-scale QCQP with $n$ decision variables can be solved by a small-scale optimizer with $\alpha_{ub}n$ decision variables. Leveraging predicted values based on hypergraph neural networks, an initial feasible solution is initially searched by dynamically adjusting the radius using our proposed *Q-Repair algorithm* (Sec. 4.2.1). Then, a neighborhood search with a fixed radius is performed with an adaptive neighborhood partition strategy (Sec. 4.2.2). Finally, for multiple sets of neighborhood search solutions, neighborhood crossover and our proposed *Q-Repair algorithm* are used to repair infeasible solutions (Sec. 4.2.3), i.e., making them feasible. The current solution is iteratively optimized using the last two steps until the time limit is reached. Finally, the current solution is output as the final optimization result.

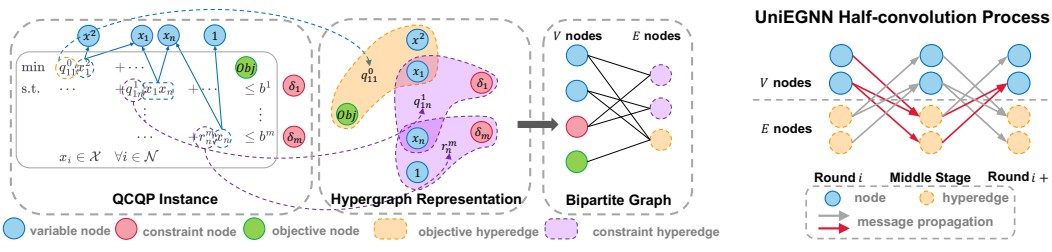

Figure 4: Variable relational hypergraph representation.          Figure 5: Half-convolution.

### 4.2.1 Q-Repair-Based Initial Feasible Solution

For a QCQP involving $n$ decision variables, NeuralQP arranges these variables in ascending order based on their predicted losses. Subsequently, it defines an initial variable proportion denoted as $\alpha$ and fixes the first $(1 - \alpha)n$ variables, leaving the remaining variables for optimization. The Q-Repair algorithm is then applied to identify constraints that are expected to be violated, and the neighborhood search scope is expanded. Throughout this process, the number of unfixed decision variables after expansion is ensured to remain below or equal to $\alpha_{ub}n$. Therefore, a small-scale optimizer can be used to effectively find the initial feasible solution.

$$
\begin{aligned}
\min \quad & \boldsymbol{x}^{\mathrm{T}}\boldsymbol{Q}^0\boldsymbol{x} + \left(\boldsymbol{r}^0\right)^{\mathrm{T}}\boldsymbol{x} \\
\text{s.t.} \quad & \boldsymbol{x}^{\mathrm{T}}\boldsymbol{Q}^i\boldsymbol{x} + \left(\boldsymbol{r}^i\right)^{\mathrm{T}}\boldsymbol{x} \le b_i \\
& l_j \le x_j \le u_j \\
& x_j = \hat{x}_j, \quad \forall x_j \in \mathcal{F} \\
& \forall i \in \mathcal{M}, \quad \forall j \in \mathcal{N}
\end{aligned}
\qquad
\xrightarrow[u_{jk}, l_{jk} = \mathrm{McCor}(u_j, u_k, l_j, l_k)]{\phi_{jk} := x_j x_k, \forall j, k \in \mathcal{N}}
\qquad
\begin{aligned}
\min \quad & \boldsymbol{x}^{\mathrm{T}}\boldsymbol{Q}^0\boldsymbol{x} + \left(\boldsymbol{r}^0\right)^{\mathrm{T}}\boldsymbol{x} \\
\text{s.t.} \quad & \sum q^i_{jk}\phi_{jk} + \left(\boldsymbol{r}^i\right)^{\mathrm{T}}\boldsymbol{x} \le b_i \\
& l_j \le x_j \le u_j, l_{jk} \le \phi_{jk} \le u_{jk} \\
& x_j = \hat{x}_j, \phi_{jk} = \hat{x}_j x_k, \forall x_j \in \mathcal{F} \\
& \phi_{jk} = \hat{x}_j \hat{x}_k, \quad \forall x_j, x_k \in \mathcal{F} \\
& \forall i \in \mathcal{M}, \quad \forall j, k \in \mathcal{N}.
\end{aligned}
$$

$$(6)$$

The Q-Repair algorithm is based on the term-wise McCormick relaxation approach (Sec. 2.3). It estimates the bounds of the left-hand side terms of the constraint conditions to quickly identify the violated constraints. First, the quadratic terms within the constraints are linearized through McCormick relaxation, which is shown in Equation 6, where $\hat{x}_j$ denotes the current solution of $x_i$ and $\mathcal{F}$ denotes the set of fixed variables. Then the linear Repair algorithm (Ye et al., 2023b) is employed. For unfixed variables, their bounds are determined based on their original problem bounds. In contrast, for fixed variables, their current solution values are utilized as the bounds for the respective terms. These bounds are all summed together to calculate the bounds for the left side of the constraint and be compared with the right-side coefficient. If the constraint is violated, the fixed variables within it are added to the neighborhood one by one until the solution becomes feasible. The related pseudocode is shown in Appendix B.1.

The Q-Repair algorithm reintroduces critical variables into the neighborhood, enabling the progressive updating of the neighborhood search radius and the repair of infeasible solutions. Because of the Q-Repair algorithm, for the first time, neighborhood crossover has been introduced into quadratic problems, thereby enhancing both solution speed and solution quality.

### 4.2.2 Neighborhood Partition

To reduce the number of violated constraints, neighborhood partitioning for QCQPs is carried out using the ACP framework (Ye et al., 2023a). After randomly shuffling the order of constraints, NeuralQP adds each variable of a constraint to a particular neighborhood one by one until the size of that neighborhood reaches its upper limit. Due to the fixed radius of the neighborhoods, the number of neighborhoods is adaptive. More details and the related pseudocode are shown in Appendix B.2.

The ACP framework for neighborhood partitioning increases the likelihood that variables within the same constraint are assigned to the same neighborhood, reducing the probability of a specific constraint being infeasible. However, in QCQPs with a large number of variables within a single constraint, this approach may result in an excessive number of neighborhoods, which can reduce

solving speed. Therefore, for QCQPs with varying degrees of density, an adaptive neighborhood partitioning strategy is necessary to prevent an excessive number of neighborhoods. When the density exceeds a certain fixed threshold, a variable-based random neighborhood partitioning strategy is employed. All variables are randomly shuffled, and NeuralQP performs neighborhood partitioning sequentially, ensuring that each variable appears in only one neighborhood.

This adaptive neighborhood partitioning strategy, on the one hand, retains the advantages of the ACP framework, ensuring the feasibility of subproblems after neighborhood partitioning. On the other hand, it addresses the issue of excessive neighborhood partitioning in dense problems within the ACP framework, which leads to longer solving times, thus ensuring solving speed.

### 4.2.3 Neighborhood Search and Crossover

After obtaining the initial feasible solution and neighborhood partitioning, NeuralQP uses small-scale solvers to perform fixed-radius neighborhood searches within multiple neighborhoods in parallel. Additionally, the neighborhood optimization approach has shown promising results in linear problems (Ye et al., 2023b). Therefore, neighborhood crossovers are also exploited for QCQPs to prevent getting trapped in local optima due to search radius limitations.

To be specific, after the neighborhood partitioning, $num\_n$ subproblems are solved in parallel using a small-scale solver. Then, neighborhood crossovers are performed between every two subproblems, resulting in $\lfloor num\_n/2 \rfloor$ neighborhoods to be explored. To address the potential infeasible issues caused by neighborhood crossovers, the Q-Repair algorithm in Sec. 4.2.1 is applied again to repair the neighborhoods. Finally, the $\lfloor num\_n/2 \rfloor$ neighborhoods are solved in parallel using a small-scale solver, and the best among them is selected as the result of this round of neighborhood optimization. If the time limit is not reached, the process proceeds to the next round; otherwise, the output serves as the final optimization result. All the related pseudocodes are shown in Appendix B.3.

## 5 Experiments

Experiments are performed on three distinct benchmark problems and two real-world libraries, QAPLIB (Burkard et al., 1997) and QPLIB (Furini et al., 2019). The three problems are Quadratic Multiple Knapsack Problem (QMKP)((Kellerer et al., 2004)), Quadratic Independent Set Problem (QIS), and Quadratic Vertex Covering Problem (QVC), with the latter two modified from Independent Set Problem (Coxeter, 1950) and Vertex Covering Problem (COOK, 1971). For each problem type, we generate small, medium, and large three scales of problems, on which we train 9 neural network models in total. Details about the benchmark problems and the dataset are shown in Appendix A.4 and C. Results of experiments on QAPLIB and QPLIB are presented in Appendix E.

In the testing stage, we use SCIP and Gurobi, the state-of-the-art academic and commercial solvers, as the baselines. Then our model is employed to obtain initial solutions, followed by iterative neighborhood search where the small-scale solvers are restricted to 30% and 50% of the number of variables. To ensure fair comparison, experiments are conducted in the following two aspects (as Ye et al. (2023b)). To study the effectiveness of NeuralQP, the objective value is compared with Gurobi and SCIP within the same wall-clock time (Sec. 5.1); to verify the efficiency of NeuralQP, the running time till reaching the same objective value is compared with Gurobi and SCIP (Sec. 5.2). More details of the datasets and additional experimental settings are listed in the Appendix C. The code for reproducing the experiment results will be open-source upon the acceptance of our paper.

### 5.1 Comparison of Objective Value

To validate the effectiveness of the proposed framework for solving large-scale QCQPs using small-scale solvers, this section compares our framework with the baseline large-scale solvers Gurobi and SCIP, evaluating their performance under the same running time. As shown in Table 1, the scales of the solver used NeuralQP are constrained to 30%and 50% of the number of variables. Compared to Gurobi or SCIP, using a solver with a scale of only 30%of the number of variables, NeuralQP can obtain better objective value compared to large-scale solvers at the same time on all benchmark problems. When the scale increases to 50%, this advantage is even more pronounced.

Table 1: Comparison of objective values with SCIP and Gurobi within the same running time. Ours-30%S and Ours-30%G mean the scale-limited versions of SCIP and Gurobi respectively, with the variable proportion $\alpha$ limited to 30%. Subscripts 1, 2, and 3 represent small-, medium-, and large-scale problems, respectively. "↑" means the result is better than or equal to the baseline. Each value is averaged among 3 similar instances.

| | $QMKP_1$ | $QMKP_2$ | $QMKP_3$ | $QIS_1$ | $QIS_2$ | $QIS_3$ | $QVC_1$ | $QVC_2$ | $QVC_3$ |
|---|---|---|---|---|---|---|---|---|---|
| SCIP | 1259.15 | 5030.62 | 10907.46 | 174.61 | 181.53 | 732.13 | 29688 | 35659 | 48348 |
| Ours-30%S | 1665.30↑ | 6052.66↑ | **14747.62**↑ | 273.95↑ | 403.46↑ | 1495.07↑ | **27274**↑ | 32816↑ | **44716**↑ |
| Ours-50%S | **1710.94**↑ | **6156.98**↑ | 14473.05↑ | **280.34**↑ | **409.75**↑ | **1509.22**↑ | 27476↑ | **32777**↑ | 44940↑ |
| Time | 25s | 200s | 1000s | 40s | 130s | 1200s | 30s | 400s | 2200s |
| Gurobi | 1672.22 | 6027.79 | 11908.34 | 278.73 | 399.89 | 1459.21 | 29740 | 34867 | 45753 |
| Ours-30%G | **1951.92**↑ | 6332.49↑ | 15127.66↑ | 280.33↑ | 410.72↑ | 1514.99↑ | **28417**↑ | 33305↑ | 45355↑ |
| Ours-50%G | 1800.37↑ | **6502.99**↑ | **15200.67**↑ | **285.07**↑ | **415.96**↑ | **1520.97**↑ | 28604↑ | **32875**↑ | **44780**↑ |
| Time | 30s | 250s | 1500s | 50s | 300s | 1200s | 25s | 300s | 2200s |

Table 2: Comparsion of running time with SCIP and Gurobi till the same objective value. Notations are similar to Table 1. $>$ indicates the inability to achieve the target objective function within the given time limit. Each value is averaged among 3 similar instances.

| | $QMKP_1$ | $QMKP_2$ | $QMKP_3$ | $QIS_1$ | $QIS_2$ | $QIS_3$ | $QVC_1$ | $QVC_2$ | $QVC_3$ |
|---|---|---|---|---|---|---|---|---|---|
| SCIP | 43.07s | 797.67s | >30000s | 2737.8s | >20000s | >30000s | 98.5s | >10000s | >18000s |
| Ours-30%S | 32.28s | 178.6s | **893.04s** | 41.98s | 405.75s | 1426.88s | **13.09s** | 398s | 1023.71s |
| Ours-50%S | **23.06s** | **106.67s** | 1235.09s | **24.20s** | **136.05s** | **865.28s** | 28.26s | **105.63s** | **2200s** |
| Objective | 1665.30 | 6052.66 | 14747.62 | 273.95 | 403.46 | 1495.07 | 27476 | 32816 | 44940 |
| Gurobi | 45.00s | 443.46s | 5089.63s | 45.00s | 1824.48s | 13247.26s | 21.46s | 7633.07s | >18000s |
| Ours-30%G | **20.15s** | 281.31s | 1123.64s | 40.35s | 405.75s | 749.21s | 4.9s | 833.94s | 4142.59s |
| Ours-50%G | 47.55s | **213.92s** | **928.74s** | **21.83s** | **136.05s** | **396.71s** | **3.55s** | **22.55s** | **1448s** |
| Objective | 1800.37 | 6332.49 | 15127.66 | 280.33 | 410.72 | 1503.99 | 29740 | 33305 | 45355 |

Furthermore, for small-scale problems, the advantages of the proposed framework compared to large-scale solvers are not very pronounced. However, when it comes to solving large-scale problems, the proposed framework significantly outperforms large-scale solvers in the same time limit. Besides, the experimental results with solvers constrained to $50\%$ scale do not consistently outperform those constrained to $30\%$ scale, indicating that small-scale solvers may be more suitable for the proposed framework, which deserves further study. All the above analyses demonstrate that the proposed NeuralQP can use small-scale solvers to effectively address large-scale QCQPs.

## 5.2 COMPARISON OF RUNNING TIME

In addition to solving performance, we also evaluate the time efficiency of NeuralQP by comparing the duration required to reach equivalent optimization results. In this aspect, both $30\%$-scale and $50\%$-scale solvers within our framework are benchmarked against their full-scale counterparts. According to the results in Table 2, NeuralQP consistently requires less time to achieve the same objective values compared to Gurobi and SCIP across all benchmark problems. Particularly for the larger-scale problems, even though the objective values achieved within a similar time limit may appear close, reaching the same optimization result typically demands substantially longer processing times for SCIP and Gurobi, often up to ten or thirty times longer, than for our proposed framework.

## 6 CONCLUSIONS

This paper presents NeuralQP, a pioneering hypergraph-based optimization framework for large-scale QCQPs. Key features of NeuralQP include: 1) a variable relational hypergraph as a complete representation of QCQPs without any assumption and 2) a McCormick relaxation-based repair algorithm that can identify illegal constraints. Experimental results demonstrate that NeuralQP achieves equivalent quality solutions in less than one-fifth of the time compared to leading solvers on large-scale QCQPs, setting the groundwork for solving nonlinear problems via machine learning. A future direction is extending our framework to nonlinear programming.

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

## APPENDIX

This Appendix contains four sections. Appendix A provides a detailed introduction to the definition and features of QCQPs and the benchmark problems used in the experiments. Appendix B shows the pseudocode and detailed explanation of *iterative neighborhood optimization*. Appendix C describes the experiment settings used in this paper, including the parameters of the experiment and the source of datasets. Appendix E presents more experimental results to further validate the effectiveness of the proposed NeuralQP in the prediction and the convergence of optimization.

## A QUADRATIC PROGRAMMING

QCQPs encompass a broad class of optimization problems, where both the objective function and the constraints are quadratic. Equation 1 lays down the foundational structure of QCQPs. In this appendix, we briefly introduce the characteristics of QCQPs, emphasizing both mathematical and machine learning-based approaches.

### A.1 CONVEXITY IN QCQPS

Convexity plays a pivotal role in the realm of optimization, significantly impacting the solvability and computational tractability of problems. A QCQP is termed convex both the objective function and the feasible region are convex. In such scenarios, global optima can be efficiently found using polynomial-time algorithms, such as interior-point methods. However, the general form of QCQPs does not have convex properties. Nonconvex QCQPs are even harder, posing significant challenges in finding global optima, and often necessitating the exploration of heuristic methods, relaxation techniques, or local search strategies.

### A.2 MIXED-INTEGER AND BINARY QUADRATIC PROGRAMS

When QCQPs incorporate integer constraints on certain variables, they evolve into Mixed-Integer Quadratically Constrained Programs (MIQCPs). MIQCPs inherit the complexities of both integer programming and quadratic programming, which combine to make them particularly challenging to solve. Binary Quadratically Constrained Programs (BQCPs), a special case of MIQCPs, impose strict restrictions on variables by exclusively allowing binary values.

### A.3 SOLUTION STRATEGIES

Solving QCQPs involves a variety of strategies. We divide these strategies into two main categories: *mathematical approaches* and *machine learning approaches*.

*Mathematical approaches* to QCQPs typically involve convex optimization algorithms for convex instances and approximation methods, branch-and-bound techniques, or metaheuristic approaches for nonconvex instances. The choice of strategy depends heavily on the problem's structure, size, and the desired balance between solution quality and computational resources. For further reference, readers are encouraged to consult key works in the field, such as Burer & Saxena (2012) and (Burer & Letchford, 2012).

The use of *machine learning methods* in solving QCQPs is notably scarce, with only a few studies, as is introduced in Sec.1. The problems in Kannan et al. (2023) are relatively small compared to our experiments. Other studies focus on Quadratic Programming (QP) problems with quadratic objective functions and linear constraints (Bonami et al. (2018), Bonami et al. (2022), Ichnowski et al. (2021) andBertsimas & Stellato (2021)), which are not directly comparable to our approach. Meanwhile, classic datasets like QPLIB (Furini et al., 2019) exist but are limited in problem count (453 in

total with 133 binary ones) and too diverse, making them unsuitable for training machine learning models due to the varied distribution of problems. Specialized datasets like QAPLIB (Burkard et al., 1997) also exist but are more focused on specific problem types.

In our method, to enable the use of small-scale solvers for general large-scale problems, we utilized three problem formulations and adopted a random generation approach. This method ensures that

1. Our approach is tested and validated across different problem types;
2. The problems generated are sufficiently large and difficult to solve.

For more details on QAPLIB and QPLIB, please refer to Appendix D.

### A.4 PROBLEM FORMULATIONS

In Section 5, we use three problem formulations: the Quadratic Independent Set Problem (QIS), the Quadratic Multiple Knapsack Problem (QKP), and the Quadratic Vertex Cover (QVC). These correspond to scenarios where *the objective function is linear with quadratic constraints*, where *the objective function is quadratic with linear constraints*, and *where both the objective function and constraints are quadratic*. Such formulation is aimed at evaluating our proposed methods in various scenarios and providing training data for testing on QPLIB.

#### A.4.1 QUADRATIC INDEPENDENT SET PROBLEM (QIS)

The Quadratic Independent Set (QIS) problem is an extension of the classical Independent Set (IS) problem (Coxeter, 1950), where we introduce quadratic non-convex constraints to enhance its relevance to realistic scenarios. In the QIS problem, we represent distinct node attributes by introducing random weights $c_i$ into the objective function, where each weight corresponds to the attribute of node $i$. To capture the high-order relationships typical in real-world networks, we employ a hypergraph model, denoted as $\mathcal{H} = (\mathcal{V}, \mathcal{E})$, where $\mathcal{V}$ is the set of vertices and $\mathcal{E}$ is the set of hyperedges. In this model, each hyperedge can encompass multiple nodes, reflecting the multi-dimensional interactions within each hyperedge. The primary objective of the QIS problem is to identify the largest weighted independent set within the graph. An independent set is a subset of vertices where each hyperedge $e \in \mathcal{E}$ satisfies certain restriction $f(e) \leq 0$. The mathematical form of the QIS problem can be expressed as:

$$
\begin{aligned}
\max \quad & \sum_{i \in V} c_i x_i, \\
\text{s.t.} \quad & f(e) \leq 0, \quad \forall e \in \mathcal{E}, \\
\text{where} \quad & f(e) = \sum_{i \in e} a_i x_i + \sum_{i,j \in e, i \neq j} q_{ij} x_i x_j - |e|, \\
& x_i \in \mathbb{B}, \quad \forall i \in V.
\end{aligned}
\tag{7}
$$

Here, the coefficients $a_i$ and $q_{ij}$ are randomly generated from a uniform distribution $U(0, 1)$. The term $|e|$ represents the degree of the hyperedge, which is chosen to ensure that the constraints are both valid and feasible.

The QIS model is highly applicable in various scenarios involving hypergraph network models with nonlinear relationships between nodes. For instance, in wireless communication networks, especially in frequency assignment tasks (Aardal et al., 2007), the challenge often involves allocating frequencies to transmitters or channels in a way that minimizes interference and maximizes network efficiency. This challenge is conceptually akin to our QIS model, necessitating consideration of nonlinear interactions between network elements. Other potential applications include social network analysis, where the intricate interplay of relationships and attributes is crucial.

#### A.4.2 QUADRATIC MULTIPLE KNAPSACK PROBLEM (QMKP)

The Quadratic Multiple Knapsack Problem (QMKP) (Hiley & Julstrom, 2006) is a combination of the Quadratic Knapsack Problem (QKP) (Gallo et al., 1980) and the Multiple Knapsack Problem

(MKP) Kellerer et al., 2004, adapted to better reflect real-world scenarios. QKP assumes a single-dimensional restriction on the items, which has limited applications; MKP assumes a linear relationship between the values of items, which ignores interactions between them. To overcome these limitations, the QMKP incorporates not only a linear value for each item but also quadratic terms representing the interactions between items when selected together, as well as multidimensional constraints. This additional complexity allows for more sophisticated modeling and is applicable in various fields. The QMKP can be mathematically formulated as follows:

$$
\begin{aligned}
\max \quad & \sum_i c_i x_i + \sum_{(i,j) \in E} q_{ij} x_i x_j, \\
\text{s.t.} \quad & a_i^k x_i \leq b^k, \quad \forall k \in M, \\
& x_i \in \mathbb{B}, \quad \forall i \in N,
\end{aligned}
\tag{8}
$$

where $c_i$ represents the value of item $i$, and the coefficient $q_{ij}$ denotes the interaction between items $i$ and $j$ when selected together. The coefficients $a_i^k$ relate to the attribute of item $i$ concerning constraint $k$, and $b^k$ is the upper limit for that attribute. In our experiment, $c_i$, $q_{ij}$ and $a_i^k$ are generated from the uniform distribution $U(0, 1)$, with $b^k = \frac{1}{2} \sum_{i \in N} a_i^k$ so that the 0s and 1s in the optimal solution are roughly the same and that the constraints remain valid.

The QMKP thus extends the scope of the QKP and the MKP, making it more suitable for scenarios where the value of items depends on the simultaneous selection of others. For example, consider optimizing a day's diet to maximize the nutritional value of each food item. In this scenario, certain food combinations may offer greater nutritional benefits, akin to the standard QKP. However, each food item also possesses various attributes such as calorie, carbohydrate, and sodium content. To ensure a healthy diet, these attributes must not exceed specific limits. The QMKP's ability to capture the interdependencies among items makes it a valuable tool for addressing complex optimization challenges in real-world settings with intricate constraints and objectives.

### A.4.3 QUADRATIC VERTEX COVER PROBLEM (QVC)

The Vertex Cover problem (COOK, 1971), a fundamental challenge in combinatorial optimization, involves selecting a minimum number of vertices in a graph such that every edge is connected to at least one selected vertex. In its classical form, the Vertex Cover problem is formulated for graph $\mathcal{G} = (V, E)$ as follows:

$$
\begin{aligned}
\min \quad & \sum_{i \in V} x_i \\
\text{s.t.} \quad & x_i + x_j \geq 1, \quad \forall (i, j) \in E, \\
& x_i \in \mathbb{B}, \quad \forall i \in V,
\end{aligned}
\tag{9}
$$

where $x_i$ is a binary decision variable indicating whether vertex $i$ is included in the vertex cover. The objective is to minimize the number of vertices in the cover while satisfying the constraints that every edge is connected to at least one chosen vertex.

However, real-world applications often require considering inter-dependencies between variables, an aspect not captured in the classical Vertex Cover formulation. To address this, we extend the problem to the QVC, introducing quadratic terms that represent the interaction between vertices. This extension allows for modeling more complex relationships and decision-making scenarios. The QVC problem is thus reformulated as follows, incorporating quadratic interactions into both the objective function and the constraints:

$$
\begin{aligned}
\min \quad & \sum_{(i,j) \in E} x_i + x_j + x_i x_j, \\
\text{s.t.} \quad & c_i x_i + c_j x_j + q_{ij} x_i x_j \geq 1, \quad \forall (i, j) \in E, \\
& x_i \in \mathbb{B}, \quad \forall i \in V,
\end{aligned}
\tag{10}
$$

where the coefficients $c_i$, $c_j$, and $q_{ij}$ represent the values and interactions between vertices $i$ and $j$. These coefficients capture the essence of the interdependencies and complex relationships inherent in many real-world scenarios, ranging from network design and logistics to social network analysis and resource allocation. In our experiment, the coefficients $c_i$, $c_j$, and $q_{ij}$ are generated from the uniform distribution $U(0,1)$.

By integrating quadratic terms in both the constraints and the objective, the QVC problem becomes one of the most challenging forms of QCQP to solve, making it a rigorous test case for any solution method. In contrast, the QIS problem involves linear objective functions with quadratic constraints, and the QMKP features quadratic objective functions with linear constraints. Our intention in formulating the QVC problem is to thoroughly evaluate our method's effectiveness across a broad spectrum of QCQP variations. The introduction of the QVC problem allows us to assess our method's performance in solving QCQPs ranging from moderately complex to highly intricate scenarios. The QVC problem, therefore, not only tests the limits of our solution strategy but also contributes to the versatility and adaptability of our approach in various domains.

# B ITERATIVE NEIGHBORHOOD OPTIMIZATION

The iterative neighborhood optimization is a crucial component of the proposed framework, enabling it to exhibit strong convergence performance and swiftly obtain high-quality feasible solutions within a short time. By applying our proposed Q-Repair algorithm, commonly used neighborhood optimization methods for linear problems can be extended to QCQPs. Iterative neighborhood optimization in this paper comprises neighborhood partitioning, neighborhood search, and neighborhood crossover. These processes are iterated sequentially to optimize the objective function value repeatedly until the time limit is reached. This section provides a detailed description of the iterative neighborhood optimization process.

## B.1 Q-REPAIR ALGORITHM

When performing fixed-radius neighborhood optimization for QCQPs, there may be some constraints that are certainly infeasible. The Q-Repair algorithm, based on the McCormick relaxation, quickly identifies these infeasible constraints and reintroduces the correlative variables into the neighborhood, thus rendering the constraints feasible once again.

The first step is using McCormick relaxation shown in Algorithm 1 to transform quadratic terms into linear terms. Then the constraints in QCQPs can be transformed into linear constraints as shown in Equation 6. Since the repair algorithm is independent of the objective function and only involves constraints, the common Repair algorithm for linear problems can be used to solve this problem after that. Assume that $Q^i_{jk} \geq 0$ and the constraint, since the case with negative coefficients can be easily generalized, the Q-repair algorithm is shown in Algorithm 2.

---
**Algorithm 1** MCCORMICK RELAXATION
---
**Input:** The upper and lower bounds of $x_1, x_2$: $u_1, u_2, l_1, l_2$
**Output:** The upper and lower bounds of $x_1 x_2$: $u_{12}, l_{12}$
  1: $u_{12} \leftarrow \min\{l_2 x_1 + u_1 x_2 - l_2 u_1, l_1 x_2 + u_2 x_1 - l_1 u_2\}$
  2: $l_{12} \leftarrow \max\{l_2 x_1 + l_1 x_2 - l_1 l_2, u_2 x_1 + u_1 x_2 - u_1 u_2\}$

---

## B.2 NEIGHBORHOOD PARTITION ALGORITHM

Neighborhood search refers to fixing a subset of decision variables to their current values and then exploring the search space for the remaining variables. The key to the effectiveness of neighborhood search is the quality of neighborhood partitioning. The neighborhood partition in the proposed framework is an adaptive partitioning based on the density of variables and the size of the neighborhood. It distinguishes the density of the problem by identifying the number of variables contained in each constraint. The number of neighborhoods is determined by the number of variables, neighborhood partition strategy, and neighborhood size.

---

**Algorithm 2** Q-REPAIR

---

**Input:** The set of fixed variables $\mathcal{F}$, the set of unfixed variables $\mathcal{U}$, the current solution $\boldsymbol{x}$, the coefficients of the given QCQP $\{\boldsymbol{Q}, \boldsymbol{r}, \boldsymbol{b}, \boldsymbol{l}, \boldsymbol{u}\}$
    $n \leftarrow$ the number of decision variables
    $m \leftarrow$ the number of constraints
**Output:** $\mathcal{F}, \mathcal{U}$

1: **for** $i \leftarrow 1$ **to** $n$ **do**
2:     **if** the i-th variable $\in \mathcal{F}$ **then**
3:         $cur\_u_i \leftarrow x_i$
4:         $cur\_l_i \leftarrow x_i$
5:     **else**
6:         $cur\_u_i \leftarrow u_i$
7:         $cur\_l_i \leftarrow l_i$
8:     **end if**
9: **end for**
10: **for** $i, j \leftarrow 1$ **to** $n$ **do**
11:     $u_{ij}, l_{ij} \leftarrow$ MCCORMICK RELAXATION$(u_i, u_j, l_i, l_j)$
12:     $cur\_u_{ij}, cur\_l_{ij} \leftarrow$ MCCORMICK RELAXATION$(cur\_u_i, cur\_u_j, cur\_l_i, cur\_l_j)$
13: **end for**
14: **for** $i \leftarrow 1$ **to** $m$ **do**
15:     $N \leftarrow 0$
16:     **for** $j \leftarrow 1$ **to** $n$ **do**
17:         $N \leftarrow N + r_j^i cur\_l_j$
18:     **end for**
19:     **for** $j, k \leftarrow 1$ **to** $n$ **do**
20:         $N \leftarrow N + Q_{jk}^i cur\_l_{jk}$
21:     **end for**
22:     **if** $N > b_i$ **then**
23:         **for** term in constraint $i$ **do**
24:             **if** is a linear term with the j-th variable $\in \mathcal{F}$ **then**
25:                 remove the j-th variable from $\mathcal{F}$
26:                 add the j-th variable into $\mathcal{U}$
27:                 $N \leftarrow N - r_j^i cur\_l_j$
28:                 $N \leftarrow N + r_j^i l_j$
29:             **else if** is a quadratic term with the j-th and k-th variable, and at least one of them $\in \mathcal{F}$ **then**
30:                 remove them from $\mathcal{F}$
31:                 add them into $\mathcal{U}$
32:                 $N \leftarrow N - Q_{jk}^i cur\_l_{jk}$
33:                 $N \leftarrow N + Q_{jk}^i l_{jk}$
34:             **end if**
35:             **if** $N \leq b_i$ **then**
36:                 BREAK
37:             **end if**
38:         **end for**
39:     **end if**
40: **end for**

---

For dense problems, the proposed framework employs a variable-based random neighborhood partition strategy. The decision variables are randomly shuffled and added to the neighborhood until the upper limit of the neighborhood size is reached. The number of neighborhoods is equal to the number of variables divided by the neighborhood size. This neighborhood partition strategy ensures that each variable appears in only one neighborhood. Even in dense problems, it guarantees that the solution speed is not compromised due to too many neighborhoods.

For sparse problems, the proposed framework uses an ACP-based neighborhood partition strategy (Ye et al. (2023a)). The constraints are randomly shuffled, and the variables within the constraints are sequentially added to the neighborhood until the upper limit of the neighborhood size is reached. This neighborhood partition strategy increases the likelihood of variables within the same constraint being in the same neighborhood, thereby reducing the probability of the constraint being bound to be violated. More details are shown in Algorithm 3.

---

**Algorithm 3** ACP-Based NEIGHBORHOOD PARTITION

**Input:** A QP, the number of variables $n$, the number of constraints $m$, the max size of the neighborhood $s_{\max}$
**Output:** A set of neighborhoods $\mathcal{N} = \{N_1, N_2, \dots\}$, the number of neighborhoods num_n
  1: Randomly shuffle the order of constraints.
  2: num_all $\leftarrow 0$
  3: **for** $i \leftarrow 1$ **to** $m$ **do**
  4:     Initialize $var\_list_i$
  5:     **for** $j \leftarrow 1$ **to** $n$ **do**
  6:         **if** The i-th decision variable is in constraint $i$ **then**
  7:             add $j$ into $var\_list_i$
  8:             num_all $\leftarrow$ num_all $+ 1$
  9:         **end if**
 10:     **end for**
 11: **end for**
 12: num_n $\leftarrow$ num_all $\setminus s_{\max}$
 13: Initialize num_n neighborhoods
 14: ID $\leftarrow 1$
 15: $s \leftarrow 0$
 16: **for** $i \leftarrow 1$ **to** $m$ **do**
 17:     **for** j $\in var\_list_i$ **do**
 18:         **if** $s < s_{\max}$ **then**
 19:             add the i-th decision variable into $N_{\text{ID}}$
 20:             $s \leftarrow s + 1$
 21:         **else**
 22:             add $N_{\text{ID}}$ into $\mathcal{N}$
 23:             ID $\leftarrow$ ID $+ 1$
 24:             $s \leftarrow 1$
 25:             add the i-th decision variable into $N_{\text{ID}}$
 26:         **end if**
 27:     **end for**
 28: **end for**

---

### B.3 NEIGHBORHOOD SEARCH AND CROSSOVER

To prevent the neighborhood search from getting trapped in local optimal, neighborhood crossover is required. The neighborhood crossover used in this paper takes place between two neighborhoods, and the details are provided in Algorithm B.3. Q-REPAIR() constructs a new search neighborhood to prevent the problem from becoming infeasible after neighborhood crossover. SEARCH() means using a small-scale solver to search for a better solution in a specific neighborhood.

---

**Algorithm 4** NEIGHBORHOOD CROSSOVER

---

**Input:** The decision variable set $\mathcal{X}$, neighborhoods $N_1, N_2$, neighborhood search solution $x^1, x^2$, the number of variables $n$

**Output:** Crossover solution $x'$

1: // Assume that $x^1$ is better than $x^2$
2: **for** $i \leftarrow 1$ **to** $n$ **do**
3:     **if** The i-th decision variable $\in N_1$ **then**
4:         $x'_i \leftarrow x^1_i$
5:     **else**
6:         $x'_i \leftarrow x^2_i$
7:     **end if**
8: **end for**
9: $\mathcal{F}, \mathcal{U} \leftarrow$ Q-REPAIR$(\mathcal{X}, \emptyset, x')$
10: $\mathcal{X}' \leftarrow$ SEARCH$(\mathcal{F}, \mathcal{U}, x')$

---

Integrating the above steps, the complete iterative neighborhood optimization process is shown in Algorithm 5. The first step is performing neighborhood partitioning and then solving all neighborhoods in parallel using small-scale solvers. After crossing over the neighborhoods pairwise, the best solution is selected as the initial solution for the next round of neighborhood optimization. This process is repeated until the time limit is reached.

---

**Algorithm 5** ITERATIVE NEIGHBORHOOD OPTIMIZATION

---

**Input:** Initial feasible solution $x$, the number of variables $n$, the max variable proportion $\alpha_{\text{ub}}$, time limit $t$

**Output:** Optimization solution $x$

1: $\mathcal{N}, \text{num\_n} \leftarrow$ PARTITION$(s_{\max} = \alpha_{\text{ub}} n)$
2: // Do the next step in parallel
3: **for** $\mathcal{F}_i, \mathcal{U}_i \in \mathcal{N}$ **do**
4:     $x_i \leftarrow$ SEARCH$(\mathcal{F}_i, \mathcal{U}_i, x)$
5: **end for**
6: // Do the next step in parallel
7: **for** $i \leftarrow 1$ **to** num\_n$/2$ **do**
8:     $x'_i \leftarrow$ CROSSOVER$(x_{2i-1}, x_{2i}, N_{2i-1}, N_{2i})$
9: **end for**
10: $x \leftarrow$ the best solution among $x'_i$
11: **if** Reach the time limit **then**
12:     **return** $x$
13: **else**
14:     Restart from row 1 with $x$
15: **end if**

---

## C  MAIN EXPERIMENT DETAILS

### C.1  DATASET

Since the existing dataset and learning methods couldn't meet the training and testing requirements (see Appendix A.3), we generated the three problems described in Appendix A.4 using random coefficients. The number of variables and constraints for the problems is shown in Table 3. In terms of training data, we generated 100 problem instances for small-scale problems; for medium-scale problems, we generated 50 problem instances; for large-scale problems, we generated 10 problem instances. We used Gurobi to solve the problems until the gap was less than or equal to 10%, and we considered the solution at this point as the optimal or near-optimal solution for the training set,

Table 3: The size of three common types of NP-hard QCQPs. QMKP denotes the Quadratic Knapsack Problem. QIS denotes the Quadratic Independent Set Problem. QVC denotes the Quadratic Vertex Cover Problem.

| Problem | Scale | Number of variables | Number of Constraints |
|---------|-------|---------------------|-----------------------|
| QMKP (Maximize) | $QMKP_1$ | 1000 | 5 |
|  | $QMKP_2$ | 5000 | 10 |
|  | $QMKP_3$ | 10000 | 12 ~13 |
| QIS (Maximize) | $QIS_1$ | 1000 | 803 |
|  | $QIS_2$ | 1500 | 1500 |
|  | $QIS_3$ | 5000 | 3750 |
| QVC (Minimize) | $QVC_1$ | 280~300 | 650~750 |
|  | $QVC_2$ | 440~450 | 900~1100 |
|  | $QVC_3$ | 540~550 | 1180~1280 |

since the rest of the time would be primarily spent on improving dual bounds. In the testing stage, we generated 5 problem instances for small-scale problems. For medium and large-scale problems, we generated 3 problem instances as the testing dataset.

## C.2 BASELINES

We used the currently most advanced and widely used solvers, Gurobi and SCIP, as baselines. The Gurobi version used is 10.0.2 with `Threads` set to 4, `NonConvex` set to 2 and other parameters set to default values. The SCIP version used is 4.3.0 with all default settings. The scaled-constrained versions are restricted to 30% and 50% of the number of variables in the original problem. To ensure the validity of comparative experiments, the scale-constrained versions of the corresponding solvers are used as the small-scale solvers in neighborhood optimization, and the original versions share the same parameters as the scale-constrained versions.

## C.3 EXPERIMENT SETTINGS

In the training stage, we implemented a UniEGNN model with 6 convolution layers and an MLP model with 3 layers. For each combination of problem type and scale, a UniEGNN model is trained on a machine with 4 NVIDIA Tesla V100(32G) GPUs for 100 epochs with an early-stop set to 20 epochs to prevent over-fitting. The dimensions of the initial embedding space, the hidden space, the MLP input (also the convolution output), and the final output are 16, 64, 16, and 1 respectively.

In the testing stage, experiments are conducted on a machine with 36 Intel (R) Core (TM) i9-9980 XE @ 3.00 GHz CPUs. During the initial feasible solution search phase, the small-scale solver was configured to prioritize finding feasible solutions and would return when found one feasible solution or upon reaching the time limit. In the iterative neighborhood optimization phase, the small-scale solver was set to prioritize searching for the optimal solution and would return when found the optimal solution or upon reaching the time limit. Besides, we limited the initial feasible solution search, each round of neighborhood search, and neighborhood crossover time to 20s for small-scale problems, and 100s for medium-scale and large-scale problems.

## D   EXPERIMENTS ON QPLIB AND QAPLIB

We also conducted tests on the QPLIB and QAPLIB datasets, two real-world data sources. Given that previous experimental results indicated Gurobi's superior speed compared to SCIP, we used small-scale Gurobi solvers for comparison against full-scale Gurobi solvers. The experiments on QPLIB and QAPLIB were carried out on a machine equipped with 128 Intel Xeon Platinum 8375 C @ 2.90 GHz CPUs. The method for problem selection and the experimental results are in the following.

## D.1 EXPERIMENTS ON QPLIB

QPLIB is a compilation of various mixed problems and datasets, as curated by Furini et al. (2019). The authors filtered out similar problems based on criteria such as variable types, constraint types, and the proportion of non-zero elements. As a result, the 453 problems in QPLIB cover a wide range of possible scenarios, including 133 binary problems.

### D.1.1 TRAINING DATASET CONSTRUCTION

Given that the problems in QPLIB are not ideally suited for training neural networks, we adopted a novel approach. The problems in QPLIB have fewer than 1000 variables, and their constraints and objective functions can be linear or quadratic. Therefore, we formed a mixed training dataset comprising 100 instances each of small-scale QIS, QMKP, and QVC problems. Subsequently, we used a neural network, trained on this mixed dataset, to predict optimal solutions on QPLIB problems, followed by neighborhood search optimization on the initial feasible solutions. The method of generating this mixed dataset is consistent with Appendix C.1.

### D.1.2 TESTING DATASET CONSTRUCTION

Targeting large-scale, time-consuming problems, we selected certain problems from QPLIB. Initially, we solved these problems using unrestricted Gurobi with a gap limit of 10% and a time limit of 600s. We filtered out problems that could be optimally solved within 100s and those whose objective values remained the same between 100s and 600s. This process resulted in 16 problems, with variable counts ranging from 150 to 676. We then utilized Gurobi as a small-scale solver at 30% and 50% scales, with `TimeLimit` set to 100s and `Threads` set to 4, to compare the objective values achieved at 100s.

### D.1.3 RESULTS

The experimental results are listed in Table 4. Among the 16 problems, our method outperformed Gurobi in 14 cases. In terms of the 30% and 50% solver scales, our method matched or exceeded Gurobi in 12 and 10 problems, respectively. Notably, the problems in QPLIB are relatively small-scale; our method's advantages are expected to be more pronounced with even larger-scale problems.

Table 4: Comparison of objective values with Gurobi within the same running time on QPLIB. Ours-30% and Ours-50% mean the scale-limited versions of Gurobi which limit the variable proportion $\alpha$ to 30% and 50% respectively. "↑" means the result is equal to or better than the baseline. "No." is the QPLIB number. "Sense" is the objective sense. "n_var" is the number of variables.

| No. | Sense | n_var | Objective value | | |
|---|---|---|---|---|---|
| | | | Ours-30% | Ours-50% | Gurobi |
| 2017 | min | 252 | -21544.0 | **-22984.0**↑ | -22584.0 |
| 2022 | min | 275 | -22000.5↑ | **-22716.0**↑ | -21514.5 |
| 2036 | min | 324 | -28960.0↑ | **-30480.0**↑ | -28260.0 |
| 2067 | min | 190 | **3311060.0**↑ | 3441020.0 | 3382980.0 |
| 2085 | min | 253 | 8154640.0 | **7717850.0**↑ | 7885860.0 |
| 2315 | min | 595 | **-13552.0**↑ | -22680.0 | -17952.0 |
| 2733 | min | 324 | **5358.0**↑ | **5358.0**↑ | 5376.0 |
| 2957 | min | 484 | 3616.0↑ | **3604.0**↑ | 3788.0 |
| 3347 | min | 676 | **3825111.0**↑ | 3828653.0 | 3826800.0 |
| 3402 | min | 144 | 230704.0↑ | **224416.0**↑ | 230704.0 |
| 3584 | min | 528 | **-13090.0**↑ | -18989.0 | -15525.0 |
| 3752 | min | 462 | -1075.0 | -1279.0 | **-1299.0** |
| 3841 | min | 300 | -1628.0↑ | **-1690.0**↑ | -1594.0 |
| 3860 | min | 435 | -13820.0 | -16331.0 | **-16590.0** |
| 3883 | min | 182 | **-788.0**↑ | **-788.0**↑ | -782.0 |
| 5962 | max | 150 | **6962.0**↑ | 6343.0↑ | 5786.0 |

## D.2 EXPERIMENTS ON QAPLIB

QAPLIB (Burkard et al., 1997) consists of 136 problems, ranging in size from 10 to 256 (corresponding to 100 to 66536 variables). Given the limited number of problems for each size, it is not suitable for training neural networks. The Quadratic Assignment Problem (QAP) can be modeled and solved in various ways (Loiola et al., 2007). Standard modeling as a quadratic objective function with linear constraints (Equation 11) is significantly slower when solved with general-purpose solvers like Gurobi and SCIP than with specific methods for QAP. As our goal is to solve general quadratic problems, we compared our approach with the state-of-the-art general-purpose solver Gurobi, rather than benchmarking against other specific algorithms for QAP.

### D.2.1 MATHEMATICAL FORMULATION

QAP can be formed as a quadratic program in Equation 11, which belongs to the general QCQP.

$$
\begin{aligned}
\min \quad & \sum_{i=1}^{n}\sum_{j=1}^{n}\sum_{k=1}^{n}\sum_{l=1}^{n} f_{ij} d_{kl} x_{ik} x_{jl} \\
\text{s.t.} \quad & \sum_{i=1}^{n} x_{ij} = 1, \quad j = 1, 2 \ldots n, \\
& \sum_{j=1}^{n} x_{ij} = 1, \quad i = 1, 2 \ldots n.
\end{aligned}
\tag{11}
$$

$f_{ij}$ represents the flow between facilities $i$ and $j$; $d_{kl}$ denotes the distance between locations $k$ and $l$; $x_{ik}$ is a binary variable that equals 1 if facility $i$ is assigned to location $k$ and 0 otherwise. The first summation constraint ensures that each location $j$ has exactly one facility assigned to it. The second summation constraint guarantees that each facility $i$ is assigned to exactly one location. A QAP can be encoded as two matrices $\boldsymbol{F}, \boldsymbol{D} \in \mathbb{R}^{n \times n}$ where $\boldsymbol{F} = [f_{ij}]$ is the flow matrix and $\boldsymbol{D} = [d_{kl}]$ is the distance matrix.

### D.2.2 TRAINING DATASET CONSTRUCTION

As QAPLIB is not suitable for neural network training, we opted for random generation. We generated two sizes of QAP problems, 20 and 50 (corresponding to 400 and 2500 variables), creating 50 and 20 instances for each size, respectively. Parameters in the QAP formula $f_{ij}$ and $d_{kl}$ were sampled from the uniform distribution $U(0, 1)$. We used Gurobi to solve the problems until the gap was less than or equal to 10% to generate labels, which is the same as is explained in Appendix C.1.

### D.2.3 TESTING DATASET CONSTRUCTION

Modeling the QAP problem in the QCQP format results in a complexity of $O(n^4)$, so we conducted tests on part of QAPLIB problems in standard form. We tested on 8 instances of size 20 (400 variables) and 3 instances of size 50 (2500 variables). Before modeling them as quadratic problems, we normalized the $\boldsymbol{F}$ and $\boldsymbol{D}$ matrices by dividing them by their maximum values respectively.

### D.2.4 RESULTS

The experimental results are detailed in Table D.2.4. Among the eight problems of size 20, our method surpassed Gurobi in 6 instances. For the solver scales of 30% and 50%, our approach matched or exceeded Gurobi in 5 and 6 problems, respectively. In all 3 problems of size 50, both our 30% and 50% scale solvers outperformed Gurobi. Notably, in problems of size 50, our 30% scale solver demonstrated the best performance, indicating that our small-scale solver combined with a neighborhood search framework has significant potential for handling large-scale problems.

Future directions might include 1) a detailed study of the dataset, generating a sufficient number of problems from the typical, limited problems in QPLIB for neural network training, and 2) improving the neural network by training larger models on diverse datasets, thus achieving one-shot learning on QPLIB.

Table 5: Comparison of objective values with Gurobi within the same running time on QAPLIB. Name is the instance name given by QAPLIB. Other notations are the same as Table 4. All QAP problems are *minimization* problems.

| Name | n_var | Time Limit (s) | Objective value | | |
|---|---|---|---|---|---|
| | | | Ours-30% | Ours-50% | Gurobi |
| chr20a | 20 | 100 | 3.03 | 2.84 | **2.62** |
| chr20b | 20 | 100 | 2.94↑ | **2.89↑** | 2.94 |
| chr20c | 20 | 100 | 1.75 | 1.61 | **1.54** |
| had20 | 20 | 100 | **62.93↑** | **62.93↑** | 63.33 |
| nug20 | 20 | 100 | **36.77↑** | 37.09↑ | 37.77 |
| rou20 | 20 | 100 | 75.65↑ | **75.16↑** | 77.61 |
| tai20a | 20 | 100 | 74.92↑ | **73.90↑** | 75.69 |
| tai20b | 20 | 100 | 2.80 | **2.51↑** | 2.52 |
| tai50a | 50 | 1800 | **524.18↑** | 527.47↑ | 531.52 |
| tai50b | 50 | 1800 | **31.73↑** | 32.47↑ | 32.76 |
| wil50 | 50 | 1800 | **427.04↑** | 427.16↑ | 427.67 |

# E    EXTRA EXPERIMENTS

In this section, the details of our two extra experiments regarding the two stages introduced in Sec. 4, *neural prediction evaluation* and *neighborhood optimization convergence analysis*, are presented to further validate the effectiveness and efficiency of our optimization framework.

## E.1    NEURAL PREDICTION EVALUATION

Our method is similar to the Neural Diving model proposed by Nair et al. (2021) in that we trained a neural network to directly predict solutions and is also similar to Relaxation Enforced Neighborhood Search (RENS) (Berthold, 2014) in that we obtain high-quality feasible solutions by fixing a subset of variables and solve the resulting sub-QCQP. In the following sections, a detailed description of our model is given, a new method to handle general integer and continuous variables is proposed and the visualization of neural prediction results is presented.

### E.1.1    MODEL FORMULATION

Given a QCQP instance $\mathcal{Q}$ of the form in Equation 1, the optimal solution $\mathcal{X}^*$ can be represented as a function $g$ of the instance $\mathcal{Q}$, denoted as $g(\mathcal{Q})$. The learning objective is then to approximate this function $g$ using a neural network model $g_\theta$, parameterized by $\theta$. Formally, the training dataset is constructed as $\mathcal{D} = \{(\mathcal{Q}_i, \mathcal{X}_i^*)\}_{i=1}^N$, where $\{\mathcal{Q}_i\}_{i=1}^N$ represents $N$ problem instances and $\{\mathcal{X}_i^*\}_{i=1}^N$ denotes the corresponding optimal or suboptimal solutions obtained by off-the-shelf solvers.

To measure the difference between the network's predictions and the true solutions, we employ the binary cross-entropy loss with logits, represented by BCEWithLogitLoss in PyTorch. This loss function is suitable as it accounts for the fact that the network outputs are not probabilities and applies a sigmoid function to convert them into probabilities before calculating the loss. The loss for each instance in the dataset is calculated as:

$$\mathcal{L}(\theta) = -\frac{1}{N} \sum_{i=1}^N \left[ y_i \cdot \log(\sigma(g_\theta(\mathcal{Q}_i))) + (1 - y_i) \cdot \log(1 - \sigma(g_\theta(\mathcal{Q}_i))) \right], \tag{12}$$

where $\sigma$ denotes the sigmoid function, $y_i$ is the true label indicating whether the solution is 0 or 1, and $g_\theta(\mathcal{Q}_i)$ is the raw output of the network for the given QCQP instance $\mathcal{Q}_i$. The goal during training is to minimize this loss function, effectively adjusting the parameters $\theta$ to improve the approximation of the true function $g$.

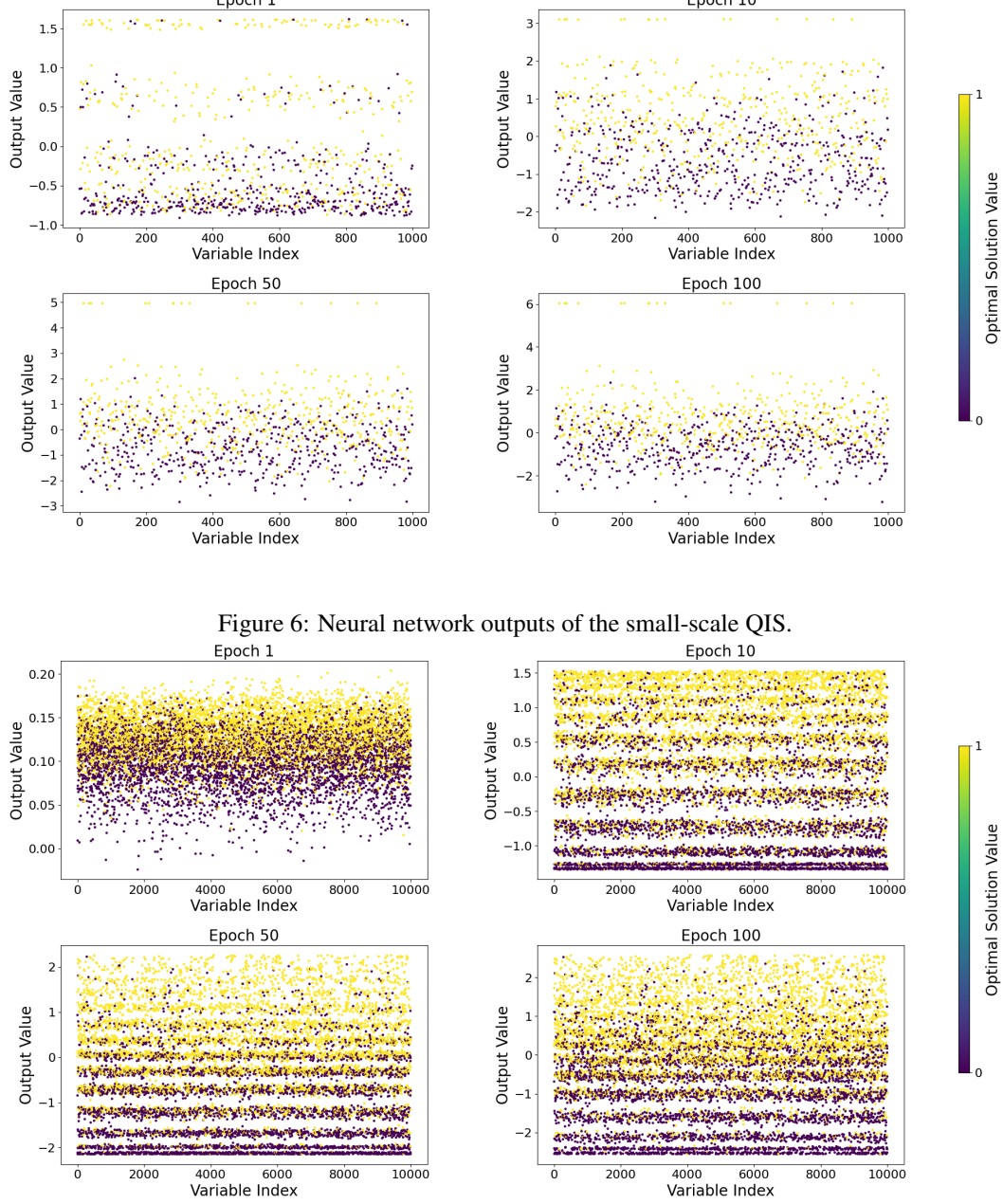

Figure 6: Neural network outputs of the small-scale QIS.

Figure 7: Neural network outputs of the large-scale QMKP.

### E.1.2 HANDLING INTEGER AND CONTINUOUS VARIABLES

For integer variables, two strategies are considered: one represents an integer as a sum of multiple binary variables, and the other, similar to the Neural Diving model (Nair et al., 2021), treats an integer in its binary form, where each bit prediction is a binary classification task. This latter approach addresses the challenges associated with variable cardinality by reframing the prediction task into a sequence of binary tasks, enabling more efficient handling of integer variables by predicting the most significant bits and subsequently narrowing the possible variable range.

Regarding continuous variables in nonconvex QCQPs, the necessity for predicted solutions arises, as not all continuous variables can be optimized during the iterative neighborhood search. This is

attributed to the requirement of spatial branching (sB&B) (McCormick, 1976) for optimizing continuous variables in nonconvex scenarios, a computationally expensive and NP-hard process. Consequently, a viable method is treating continuous variables as integer variables in training instances. This adaptation enables the model to predict high-quality integer values, serving as initial feasible solutions for sB&B in the neighborhood optimization process, thereby facilitating a balanced computational effort.

### E.1.3    PREDICTION RESULTS VISUALIZATION

Given the inherent variance in true labels (0s and 1s in binary problems) across optimal solutions of distinct QCQP instances, both loss and accuracy metrics lose some degree of meaningfulness in the evaluation of the efficacy of neural prediction. Consequently, we opt for a visualization strategy wherein we plot the output solution values of a specific QCQP instance across various epochs, facilitating the observation of a discernible trend of separation amongst the variables of differing optimal values. The $x$ axis represents the index of variables, while the $y$ axis delineates the output values emanating from the neural network model, with a larger value signifying a higher confidence level of our model predicting it as 1. A value approaching 0 indicates a degree of uncertainty within the model concerning the optimal value—a plausible scenario given that the value of certain variables might exert a negligible influence on the objective values. The neural network outputs for both small-scale QIS instances (Figure 6) and large-scale QMKP instances (Figure 7) are depicted, serving to demonstrate that our proposed network has indeed acquired the capability to discern the optimal solution values, with similar plots being observable across other problem types of varying scales. Inspection of these plots reveals instances of misclassification by the neural network, indicating a potential area for enhancement in subsequent research endeavors. Additionally, the emergence of a "stripe"-like pattern within large-scale instances is noted, suggesting the potential classification of variables within the same "stripe" into identical categories, which is a phenomenon warranting further exploration in future work.

### E.2    NEIGHBORHOOD OPTIMIZATION CONVERGENCE ANALYSIS

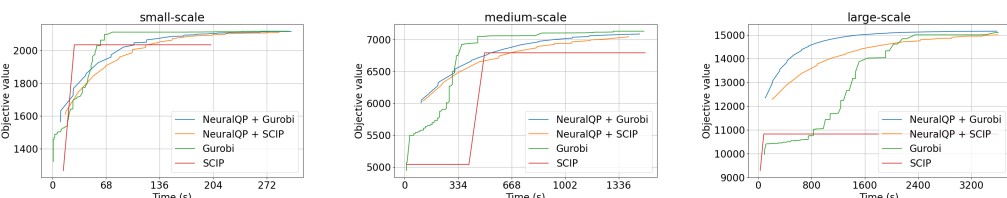

Figure 8: The time-objective value figure for QMKP.

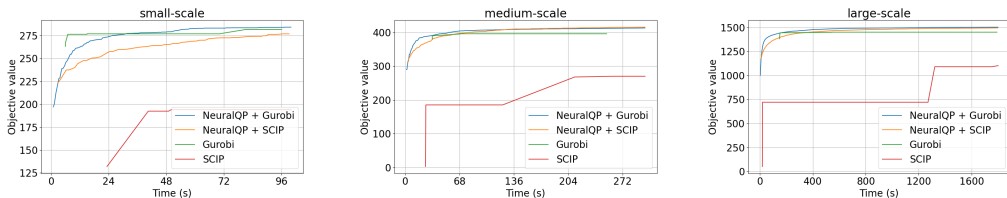

Figure 9: The time-objective value figure for QIS.

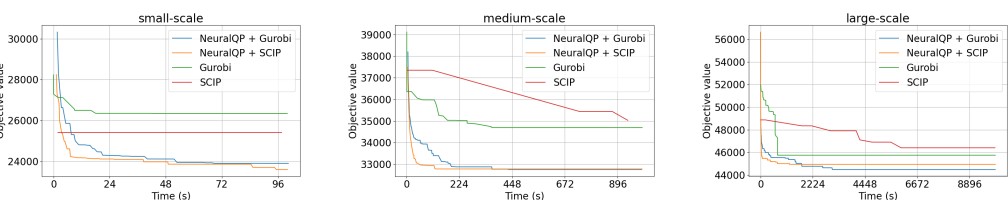

Figure 10: The time-objective value figure for QVC.

Convergence is an important criterion for assessing the effectiveness of neighborhood optimization. It can be obtained by observing the curve of the objective value over time during the optimization process. The convergence curves for solving QCQPs by using the proposed framework with the small-scale version solver, in comparison with the large-scale version solver, as shown in Figure 8, Figure 9 and Figure 10, with the small-scale solvers in our method all restricted to 30%.

It is apparent that our proposed framework exhibits superior convergence performance compared to Gurobi and SCIP, especially in large-scale problems. It converges to high-quality solutions in less time. Interestingly, despite the convergence performance of SCIP's noticeably weaker compared to Gurobi, the proposed framework with restricted SCIP as the small-scale solver outperforms the restricted Gurobi in the Quadratic Vertex Cover problem.

### E.3    AREA-UNDER-CURVE VS. RUNNING TIME

We have further plotted curves of the area-under-curve (AUC) divided by running time. In such graphs, if the objective function value converges, the function should approach a horizontal line. The quicker the function approaches this line, the faster the convergence. For maximization problems, a higher y-coordinate value of this asymptotic line is preferable; conversely, for minimization problems, a lower value is better.

#### E.3.1    EXPERIMENT RESULTS

Since our focus is on large-scale problems, we have drawn graphs for both medium-scale and large-scale scenarios. As the graphs for each type of problem at each scale exhibit similarities, we have selected one representative graph to illustrate our findings.

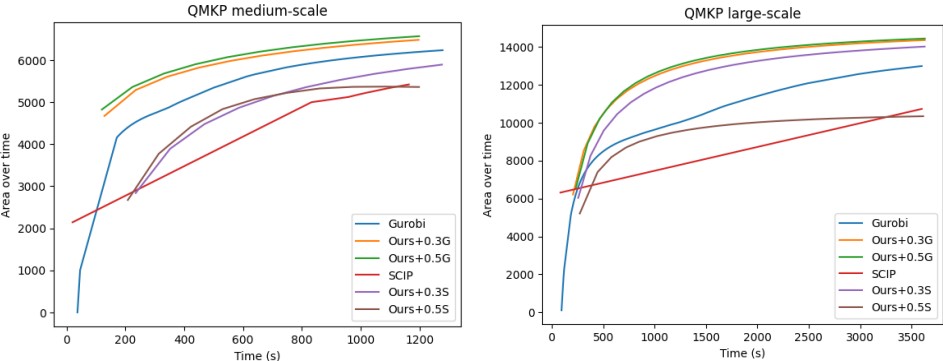

Figure 11: The time-objective value figure for QMKP.

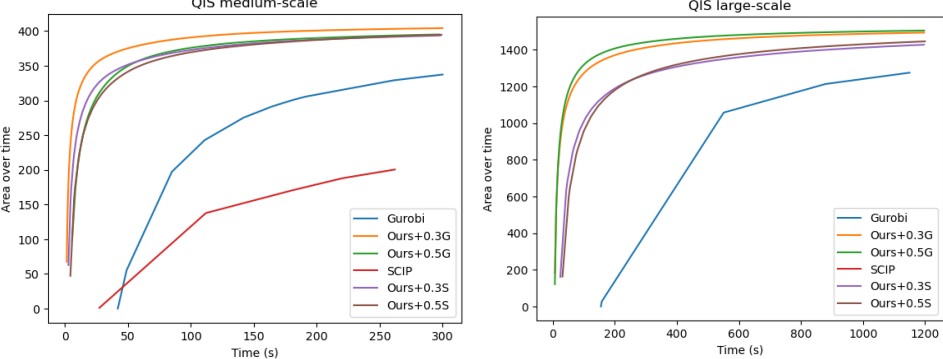

Figure 12: The time-objective value figure for QIS.

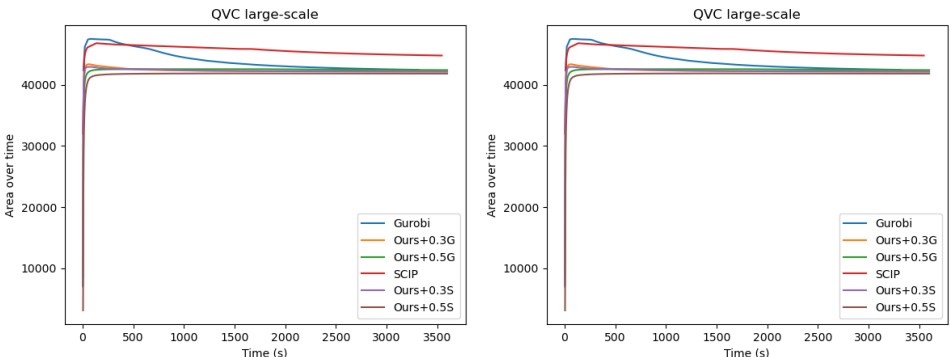

Figure 13: The time-objective value figure for QVC.

### E.3.2 RESULTS ANALYSIS

QMKP and QIS are framed as maximization problems, whereas QVC is a minimization problem. For QMKP and QIS, our analyses reveal that our method quickly increases the objective value and demonstrates faster convergence, and thus the AUC divided by time value increases sharply and converges quickly. In the case of QVC, it is observable that our approach converges to a horizontal line in a shorter time frame. In contrast, Gurobi and SCIP initially exhibit higher objective function values during the early stages of the optimization process. This results in a larger AUC divided by time, correlating with a rapid initial ascent followed by a gradual decline, as depicted in Figure 13.

Since this is the first attempt to include such a metric in the field of learning for optimization, we are cognizant that our understanding of its full implications may be limited at this stage. We wholeheartedly welcome future discussions and in-depth research on this topic.

