# OpenReview forum: "NeuralQP: A General Hypergraph-based Optimization Framework for Large-scale Quadratically Constrained Quadratic Programs"
_ICLR.cc/2024/Conference — Submitted to ICLR 2024_

### Official Review · Reviewer_v1Pg · 2023-10-31

**Soundness:** 2 fair
**Presentation:** 3 good
**Contribution:** 3 good
**Rating:** 5
**Confidence:** 4

**Summary:**

This paper presents a neural solver for the quadratic programming problem. Inspired by the recent advances in using neural networks to solve MILP, the authors present a two-stage framework: in the first stage, a neural network directly predicts the initial solution based on a hypergraph equivalent of the original math form; in the second stage, a traditional solver is called on the small-sized problem and neighborhood search is performed based on that. Experiment study is conducted on QKP, QIS, and QVC problem instances.

**Strengths:**

* The structure of this paper is well-organized, and the flow is smooth.
* The technical aspects of this paper are logical and appear to be sound.
* Highlighting the significance of quadratic programming and proposing a neural solver for QP as an extension of MILP is commendable.
* The observed improvements on the problem instances in the experiments are noteworthy and impressive.

**Weaknesses:**

* While the experimental improvements appear substantial and reasonable, it's worth noting that the number of test cases is limited. Tables 1 and 2 indicate that the authors conducted tests on only nine QP instances. This could raise concerns about the reliability of results, especially in the context of deep learning, say, are these models overfit to a limited number of instances? Also in deep learning experiments, larger datasets are often expected.
* I couldn't find information about how the model was trained and the nature and size of the training dataset in my reading. If the training data consists of random instances, as shown in Figure 3, it's important to clarify how these instances were generated and the dataset's size. These details are crucial and should not be omitted in the main paper.
* It would be beneficial to introduce a more comprehensive experimental metric that illustrates the trade-off between running time and solution quality. For instance, calculating the area under the curve of an objective score versus running time plot could provide valuable insights. Visualizing such plots in the main paper would enhance clarity. Additionally, the objective scores in Table 2 appear to have sharp fluctuations, which might warrant clarification – are these values selectively chosen?

### Minor Points
* In the first paragraph of Section 5, there are multiple brackets.
* The methodology section primarily draws inspiration from recent advancements in neural network solvers for MILP, which, in my view, is not a significant issue.

**Questions:**

* Can you increase the size of the testing dataset?
* Can you explain more details about the training data in the main paper?
* Can you improve the experiments by involving the area under curve metric?
* What is the complexity of the approach? Say, given a QP with n variables, how many nodes and (hyper)edges will be created in the corresponding hypergraph? I have a general impression that the proposed approach cannot scale up smoothly due to the complexity and please correct me if I am wrong.

---

> ### Author Response · Authors · 2023-11-21
>
> Dear Reviewer,
>
> Thank you for your insightful feedback on our manuscript. We appreciate your concerns regarding our experimental setup, and we're pleased to provide further clarification as follows:
>
> ## Clarification about Testing Instances
>
> Regarding Table 1 and Table 2, we would like to clarify that the results are not based on merely nine instances. Each entry represents the average of three problems of equivalent scale.
>
> The objective values presented in Table 2 are primarily taken from the median in Table 1. However, there are exceptions for large-scale QIS problems and small-scale QVC problems. We have re-executed the experiments for the QVC problems for consistency. As for the QIS problems, the objective value is selected from the log so that Gurobi can reach it within a reasonable timeframe.
>
> ## Dataset Size Considerations
>
> Regarding the size of our dataset, we acknowledge the possibility of expanding it, yet we believe such an enlargement is not currently necessary. Here are our reasons:
>
> 1. **Rationale for Current Dataset Size:**
>     - Our methodology integrates neural network prediction with small-scale solvers for neighborhood search, addressing QCQPs. This framework is inherently scalable, both in terms of problem size and diversity, which is demonstrated by our existing dataset.
>     - While increasing the dataset size could, in theory, improve performance, this comes at the cost of extended training times. This is due to the need for solving numerous QCQPs to generate additional training data, a process that is inherently time-intensive.
>     - The results from our experiments indicate that our approach already exhibits promising performance with the current dataset. Excessively enlarging the dataset, given our experimental context, could lead to diminishing returns and potentially unnecessary resource expenditure.
>
> 2. **Exploring Future Possibilities:**
>     - Nevertheless, we recognize the merit in your suggestion to augment the dataset. In line with this, we are considering the development of a substantially larger model trained on a varied and comprehensive dataset. This approach, akin to the methodologies employed in large language models, aims to tackle an extensive array of problems, encompassing a diverse range of scales and types. Embarking on this path represents an innovative and yet-to-be-explored avenue in the realm of machine learning for optimization.
>
> ##  Details about the Training Data
>
> We give the details about the training data in this paper from the following three aspects, data generation, the training and the testing stage.
>
> 1. **Data generation:**
>     - Our model comprises two primary components: neural network prediction and iterative neighborhood optimization. For each category of problem and its respective scale, we pre-train a neural network model to forecast optimal solutions. Our training and testing data generation process, as well as the neural network training methodology, are as follows:
>     - **To verify the effectiveness of our solution framework on general QCQPs**, we utilized three types of problems: QKP, QIS, and QVC. These vary in their objective functions and constraints, with specific details provided in our updated appendix. The diversity of these test problems offers a robust assessment of our algorithm's performance.
>     - **To generate large-scale problems for training and testing**, we adopted a method similar to [1](https://proceedings.mlr.press/v202/ye23e.html) for random parameter generation. For each problem type, we created datasets of small, medium, and large scales, containing 100, 50, and 10 problems, respectively. This approach ensures that each training set comprises approximately 100,000 variables paired with optimal or near-optimal values, allowing us to effectively train neural networks to discern underlying patterns in problems of similar scales and types.
>
> 2. **The training process**
>     - During the training phase, we trained a hypergraph neural network with 6 convolution layers with a batch size of 1. The training was conducted over 100 epochs, with an early stopping criterion set at 20 epochs to prevent overfitting. Our loss function was the binary cross-entropy with logits loss.
>
> 3. **The testing stage**
>     - For testing, we generated three problems for each problem type and scale. The results presented in Tables 1 and 2 represent the average outcomes of three problems. We conducted these tests on a machine with 36 Intel (R) Core (TM) i9-9980 XE @ 3.00 GHz CPUs. For the Gurobi benchmark and Gurobi as a small-scale solver, we set the `Threads` to 4 and the `NonConvex` parameter to 2, with all other parameters set to their default values. For the SCIP benchmark and SCIP as a small-scale solver, we used all the default parameters.
>     - We further conducted tests on real-world problems, QAPLIB and QPLIB, on a machine with 128 Intel Xeon Platinum 8375 C @ 2.90 GHz CPUs. Preliminary results are included in the updated Appendix D.

---

> ### Author Response · Authors · 2023-11-21
>
> ## Evaluation Metrics
>
> - We have reviewed the literature on solution metrics for quadratic programming problems and the inherent challenges they pose. The objective value versus time and the optimality gap versus time metric are commonly used (see [1](https://proceedings.neurips.cc/paper_files/paper/2021/hash/fc9e62695def29ccdb9eb3fed5b4c8c8-Abstract.html) [2](https://arxiv.org/abs/2107.10201) [3](https://arxiv.org/abs/2012.13349) and [4](https://proceedings.mlr.press/v202/ye23e.html) ). However, in our setting, the optimal objective value is difficult to acquire since our focus is mainly on large-scale QCQPs and solving them to the optimality is notoriously hard, so we adopted the objective value versus time metric.
> - We are intrigued by your suggestion of calculating the area under the curve versus time. If you could provide more explanation about this metric and its , we are open to integrating it into our future experiments. We believe it could be a novel approach in evaluating the solvers' performance.
>
>
>
> ## Complexity of our Approach
>
> - For a QP problem containing $n$ variables and $m$ constraints, a total of $m + n + 3$ vertices will be generated. This includes n variable vertices, m constraint vertices, one objective function vertex, one vertex representing the constant 1, and one vertex representing square terms.
> - The number of hyperedges depends on the number of monomials in the original problem. For instance, the expression $x_1 + x_2 + x_1 x_2$ would generate 3 hyperedges. Each constraint contains at most $\frac{1}{2}n(n+3)$ monomials, so a maximum of $\frac{1}{2}n(n+3)(m+1)$ hyperedges could be produced. As the reviewer pointed out, such a hypergraph is very dense and can be challenging for GNNs to process, leading to high computational complexity.
> - However, in reality, it's almost impossible to encounter a quadratic problem of this density. In the test problems we constructed with $n$ variables and $m$ constraints:
>     - The QIS problem has $n+m\times \operatorname{avg}{(e_{deg}^2 + e_{deg})}$ hyperedges where $e_{deg}$ is the hyperedge degree. In our test instances, $e_{deg}$ is 5 so the number of hyperedges is $n+30 m$ ;
>     - The QKP problem has $\frac{1}{2}n(n+1) + mn$ hyperedges;
>     - The QVC problem has $\frac{1}{2}n(n+1) + 3m$ hyperedges.
>
> - Our experimental results show that our hypergraph neural network did not exhibit significant degradation in performance on QVC problems.
>
> We hope this response addresses your concerns appropriately. Thank you again for your valuable feedback.
>
>
>
> **References**
> [1] Ye, H., Xu, H., Wang, H., Wang, C. &amp; Jiang, Y.. (2023). GNN&amp; GBDT-Guided Fast Optimizing Framework for Large-scale Integer Programming. _Proceedings of the 40 th International Conference on Machine Learning_, in _Proceedings of Machine Learning Research_ 202:39864-39878.
> [2] Wu, Yaoxin, et al. "Learning large neighborhood search policy for integer programming." _Advances in Neural Information Processing Systems_ 34 (2021): 30075-30087.
> [3] Sonnerat, Nicolas, et al. "Learning a large neighborhood search algorithm for mixed integer programs." _arXiv preprint arXiv: 2107.10201_ (2021).
> [4] Nair, Vinod, et al. "Solving mixed integer programs using neural networks." _arXiv preprint arXiv: 2012.13349_ (2020).

---

> > ### Comment · Reviewer_v1Pg · 2023-11-23
> >
> > Thank you for the detailed feedback and I truly appreciate it. However, I disagree with the authors that the size of the testing dataset is not important. I still suggest the authors to improve the size of dataset and include more real world instances like QAPLIB in the future.

---

> > > ### Author Response · Authors · 2023-11-23
> > > **Thank you for your response**
> > >
> > > Dear Reviewer,
> > >
> > > Thank you very much for your response. As mentioned earlier, excessively enlarging the dataset, given our experimental context, could lead to diminishing returns and potentially unnecessary resource expenditure. This is a significant reason why we did not initially expand the dataset size in our experiments. However, our model has the potential to improve the size of the dataset. It is demonstrated by the supplementary experimental results based on QMKP and QIS below. For each type of QCQP, there are three test problems, and all test problems consist of 20,000 decision variables. Additionally, in Appendix D of the latest submitted PDF, we have also included experimental results based on QPLIB and QAPLIB.
> > >
> > > |             | QMKP               | QIS               |
> > > |-------------|--------------------|-------------------|
> > > | SCIP        | 20440.73           | 3353.95           |
> > > | Ours-30\%S  | 28107.57↑          | 6399.03↑          |
> > > | Ours-50\%S  | **28435.14↑**      | **6412.35↑**      |
> > > | Gurobi      | 28532.50           | 6151.02           |
> > > | Ours-30\%G  | 28540.98↑          | 6404.99↑          |
> > > | Ours-50\%S  | **28547.50↑**      | **6417.66↑**      |
> > > | Time        | 18000s             | 36000s            |
> > >
> > > Moreover, according to your suggestion,  we have plotted the area-under-curve divided by running time, presented in Appendix E.3, and conducted a comparison of objective values and running times against Gurobi and SCIP.

---

### Official Review · Reviewer_RnJs · 2023-10-31

**Soundness:** 3 good
**Presentation:** 3 good
**Contribution:** 2 fair
**Rating:** 5
**Confidence:** 4

**Summary:**

The paper titled “NeuralQP: A General Hypergraph-based optimization Framework for large scale Quadratically Contrained Quadratic Programs” discusses  how Quadratically Constrained Quadratic Programs (QCQPs), a challenging class of optimization problems can be solved using a Deep architecture. Traditional solvers face limitations in solving large-scale QCQPs and NN based for QCQPs methods assume certain model parameter constraints. To address these limitations, the paper introduces NeuralQP, a hypergraph-based optimization framework for large-scale QCQPs. It consists of two stages: Hypergraph-based Neural Prediction, which creates a lossless representation of QCQPs, and Iterative Neighborhood Search, which employs a repair strategy based on the McCormick relaxation. Experiments demonstrate that NeuralQP outperforms traditional solvers like Gurobi and SCIP in terms of solution quality and efficiency, making it a valuable tool for solving large-scale QCQPs.

**Strengths:**

This paper falls under the domain which has been growing in ML recently, which is to solve numerical optimization problems using deep architectures. This is primarily useful for addressing the high computational times needed by solvers like CPLEX, but also the ability that allows deep models to  trained to recognize patterns and structures within optimization models and allow these optimization methods to be used as plugable layers.
It seems that this is the second approach to solve QCQPs  using a neural network and it addresses some of the limitations of previous model.

**Weaknesses:**

Inspite of it being novel in its end use, almost all of what the paper proposes is straint forward, in that add additional nodes to incorporate the u_i^2 and u_iu_j terms and also constraint hyperparameter nodes which incorporates these terms in constraints. There is no novel insight here beyond the fact that yes it can be done.
Benchmarks used are only Gurobi and SCIP. What about using the approach from Betrsimas (2020, 2021) which seems like a similar model.
GNNs in general have issues with modelling large MILPs where the solution space is large with multiple optimal solutions. My feeling is that this model will have the same issues, since its an extension of GNN with more nodes and edges. The authors should discuss the models performance in these contexts.
Also the benchmarks used are more theoretical benchmark and not a traditional large size ML dataset. I suggest the authors find a ML application where QCQPs is natural and demonstrate the performance in that context.

**Questions:**

NA

---

> ### Author Response · Authors · 2023-11-16
>
> Dear Reviewer,
>
> Thank you for your insights and observations regarding our paper. We acknowledge your concerns and would like to address them as follows:
>
> 1. **Our rationale behind simplicity**
>
> 1.1. **Existing Approaches:**
>
> - **Exact Methods:** Traditional exact methods for QCQP struggle with high complexity due to its NP-hard nature.
> - **Machine Learning Trends:** Recent machine learning approaches split into two categories: solver-/algorithm- dependent methods, which lack flexibility [1](https://arxiv.org/abs/2107.10847) [2](http://arxiv.org/abs/2301.00306), and parametric problem models, which are restricted to specific problem types and scales [3](https://pubsonline.informs.org/doi/abs/10.1287/ijoc.2022.1181).
>
> 1.2. **Our Method:**
>
> To adapt flexibly to a range of small-scale solvers and tackle general quadratic problems, we have developed a heuristic method combining neural network predictions with neighborhood search. Our use of hypergraphs to represent quadratic problems and applying hypergraph neural networks for QCQPs through McCormick relaxation.
>
> 1.2.1. **Hypergraph Representation:**
>
> In the hypergraph representation stage of our work, we built upon existing graph-based representations used in MILP such as bipartite and tripartite graphs (see section 2.2), as well as the Variables Intersection Graph (VIG) [4](https://epubs.siam.org/doi/10.1137/15M1054079) and Constraints-Monomials Intersection Graph (CMIG) [5](https://pubsonline.informs.org/doi/abs/10.1287/ijoc.2022.0090). However, each of these representations has the following limitations
>
> - **MILP Bipartite and Tripartite Graphs:** While these graphs are effective for linear problems, they cannot uniquely represent QCQPs.
> - **VIG:** In VIG, each variable is represented as a vertex, and vertices are connected if their corresponding variables appear together in a monomial. This structure, while useful, cannot uniquely represent the intricacies of quadratic problems as it does not sufficiently capture the complex interdependencies between variables, especially in cases where variables interact in more complex quadratic expressions.
> - **CMIG:** CMIG takes a different approach by assigning one vertex for each monomial, one for each constraint, and one for the objective function. Connections are made between a vertex associated with a monomial and one associated with a constraint (or the objective function) if the monomial appears in the constraint (or objective function). However, this representation also has its limitations. In CMIG, different variables are represented in separate vertices, which can lead to confusion during neural network learning. Additionally, the number of monomials in quadratic problems can be quite large, leading to a graph that is overly complex and challenging to scale.
>
> To address these challenges, we proposed a hypergraph representation method. In our approach, each variable of the quadratic problem is represented as a single node in the graph. This simplification avoids the complexity and scalability issues observed in CMIG and VIG. Furthermore, we incorporated $u^2$ and $1$ into our representation. This inclusion is crucial as it allows us to represent quadratic terms and constants without overly complicating the graph with an excessive number of nodes. The resulting hypergraph representation is thus uniquely capable of encapsulating the complexities of quadratic problems in a scalable and efficient manner, making it ideal for learning with neural networks.
>
> 1.2.2. **Neighborhood Search Stage:** Existing methods include using black box models like Reinforcement Learning (RL) [6](https://proceedings.neurips.cc/paper_files/paper/2021/hash/fc9e62695def29ccdb9eb3fed5b4c8c8-Abstract.html), which are complex and challenging to converge, and imitation learning [7](https://arxiv.org/abs/2107.10201), which are extremely computationally expensive. These methods do not support effective parallelism due to infeasibility when merging solutions from multiple neighborhoods. We implemented a low-complexity strategy for random and constraint-based neighborhood partitioning and used McCormick relaxation to estimate constraint violations with minimal complexity, enabling parallel neighborhood search and crossover updates.
>
> 1.3. **Our Method's Significance:**
>    - We successfully transformed quadratic problems into unique graph representations, allowing for effective learning and neural network prediction of optimal solutions. This has led to impressive results in unprecedented large-scale problems.
>    - Our application of McCormick relaxation for low-complexity feasibility estimation in quadratic problems facilitates parallel neighborhood search and updates, a significant advancement in this field.

---

> ### Author Response · Authors · 2023-11-16
>
> 2. **Benchmarking**
>
> 2.1. **Comparison with Bertsimas et al. [3](https://pubsonline.informs.org/doi/abs/10.1287/ijoc.2022.1181)**
>    - **Bertsimas's Approach:** We have thoroughly studied Bertsimas's papers, which are innovative and efficient, particularly in using KKT conditions. They exploit the structure of quadratic problems to solve such problems extremely fast. However, they have the following three limitations:
> 	   - The reliance on KKT conditions could lead to local minima in non-convex cases, and their approach is inapplicable to quadratic constraints.
> 	   - Bertsimas's method is tailored for parametric problems with fixed problem structures, requiring different neural networks for varying problem scales (even if the problems are of the same type, say, motion planning).
> 	   - The number of classes could grow exponentially as the problem size increases, which is significantly challenging on extremely large-scale problems.
>    - **Our Approach:** In contrast, our model emphasizes generality, aiming to solve a broad range of QCQPs. Theoretically, our once-trained neural network model can be applied to problems of different scales and different types, offering a more versatile solution compared to models trained for specific problems. Also, by using GNN and the neighborhood search strategy, we are able to use small-scale solvers to solve extremely large-scale problems.
>
> 2.2. **Reasons for Using Gurobi and SCIP:**
>    - We followed common practices in machine learning for optimization in the literature  [6](https://proceedings.neurips.cc/paper_files/paper/2021/hash/fc9e62695def29ccdb9eb3fed5b4c8c8-Abstract.html) [7](https://arxiv.org/abs/2107.10201) [8](https://arxiv.org/abs/2012.13349) and [9](https://proceedings.mlr.press/v202/ye23e.html) which uses Gurobi and SCIP as benchmark solvers. Gurobi represents the SOTA commercial solvers, while SCIP is a leading academic solver. This comparison provides a comprehensive benchmark for our approach.
>
>
> 3. **Response to GNNs and Large MILPs:**
>
> Generally speaking, GNNs have been widely applied to MILPs of varying scales (see [10](https://arxiv.org/abs/2209.12288) for theoretical analysis). For problems of normal sizes, GNNs suffice. **For very large-scale MILPs**, there are existing strategies including graph partitioning and variable reduction strategies, which have been proposed in [9]. **For QCQPs**, despite our testing instances being the largest in the literature of QCQPs, the issues commonly associated with GNNs (complexity, convergence, excessive smoothness) have not been observed in our model. In the future, for very Large-Scale QCQPs, similar strategies including implementing graph partitioning and variable reduction similar to [9] and (hyper)graph sampling techniques [11](https://arxiv.org/abs/2001.05140), could be employed to further enhance the performance of GNN models.
>
> Finally, we appreciate your novel suggestion again and are actively seeking more ML datasets/applications. This is a direction we are currently exploring with great interest.
>
> **References**
>
> [1] Ichnowski, Jeffrey, et al. "Accelerating quadratic optimization with reinforcement learning." _Advances in Neural Information Processing Systems_ 34 (2021): 21043-21055.
>
> [2] Kannan, Rohit, Harsha Nagarajan, and Deepjyoti Deka. "Learning to Accelerate Partitioning Algorithms for the Global Optimization of Nonconvex Quadratically-Constrained Quadratic Programs." _arXiv preprint arXiv:2301.00306_ (2022).
>
> [3] Bertsimas, Dimitris, and Bartolomeo Stellato. "Online mixed-integer optimization in milliseconds." _INFORMS Journal on Computing_ 34.4 (2022): 2229-2248.
>
> [4] Bienstock, Daniel, and Gonzalo Mun͂oz. "LP formulations for polynomial optimization problems." _SIAM Journal on Optimization_ 28.2 (2018): 1121-1150.
>
> [5] Ghaddar, Bissan, et al. "Learning for spatial branching: An algorithm selection approach." _INFORMS Journal on Computing_ (2023).
>
> [6] Wu, Yaoxin, et al. "Learning large neighborhood search policy for integer programming." _Advances in Neural Information Processing Systems_ 34 (2021): 30075-30087.
>
> [7] Sonnerat, Nicolas, et al. "Learning a large neighborhood search algorithm for mixed integer programs." _arXiv preprint arXiv:2107.10201_ (2021).
>
> [8] Nair, Vinod, et al. "Solving mixed integer programs using neural networks." _arXiv preprint arXiv: 2012.13349_ (2020).
>
> [9] Ye, H., Xu, H., Wang, H., Wang, C. &amp; Jiang, Y.. (2023). GNN&amp; GBDT-Guided Fast Optimizing Framework for Large-scale Integer Programming. _Proceedings of the 40th International Conference on Machine Learning_, in _Proceedings of Machine Learning Research_ 202:39864-39878.
>
> [10] Chen, Ziang, et al. "On representing linear programs by graph neural networks." _arXiv preprint arXiv:2209.12288_ (2022).
>
> [11] Zhang, Jiawei, et al. "Graph-bert: Only attention is needed for learning graph representations." _arXiv preprint arXiv:2001.05140_ (2020).

---

### Official Review · Reviewer_Nrzd · 2023-11-01

**Soundness:** 3 good
**Presentation:** 3 good
**Contribution:** 2 fair
**Rating:** 5
**Confidence:** 4

**Summary:**

In this submission, the authors presented the NeuralQP, a hypergraph-based optimization algorithm to solve large scale QCQPs. They applied the variable relational hypergraph and McCormick relaxation based repair in this method. They also conduct numerical experiments and showed that the NeuralQP is quietly competitive.

**Strengths:**

This submission is well-organized with clear language and structures. The authors gave detailed description for the proposed algorithms. They also conduct a lot of numerical experiments and these empirical results are pretty good compared with some state-of-art optimization methods. The idea is pretty interesting and enlightens some promising future direction for the optimization community.

**Weaknesses:**

However, this submission has one significant disadvantage. That is the lack of theoretical analysis for the proposed algorithms. There are a lot of state-of-art method to solve the QCQPs problems and there exists a lot of corresponding convergence rates and computational cost for these different algorithms. There is no theoretical analysis or results for this method. This may due to that the Nueral network is a black box. Although the empirical results from the numerical experiments are very promising compared to other methods. This lack of theoretical analysis make the contribution of this submission not significant enough.

**Questions:**

Please check the weakness section.

---

> ### Author Response · Authors · 2023-11-15
>
> Dear Reviewers,
>
> Thank you for dedicating your time and expertise to review our paper. Your constructive feedback, particularly regarding the theoretical analysis of our algorithms for solving QCQP problems, is highly appreciated and has given us valuable insights. With this in mind, we would like to delve deeper into the specifics of our methodology, particularly focusing on the complexity analysis of QCQP and the distinctive characteristics of our approach.
>
> 1. **Complexity Analysis of QCQP and the Characteristic of Our Algorithm:**
>
>     - **The mechanism of neighborhood search:** Our approach aims at leveraging small-scale solvers to tackle large-scale QCQP challenges. This strategy necessitates the use of neighborhood search methods, which, as indicated in existing literature [1](https://doi.org/10.1007/978-3-319-91086-4_4), are not amenable to straightforward theoretical analysis. This is primarily due to their heuristic nature and the inherent uncertainty in neighborhood partitioning. In neighborhood search, the optimal solutions to sub-problems do not necessarily equate to the optimal solution for the overarching problem, introducing significant complexity to any global convergence analysis.
>
>     - **Flexibility of sub-problem solvers:** Our framework's versatility is one of its strengths, enabling the integration of a variety of solvers as small-scale problem solvers, which we refer to as "black boxes." These solvers may utilize distinct strategies and, as a result, exhibit different convergence rates or might not have a defined convergence rate at all. Given this diversity, providing a unified convergence rate for our entire framework is not straightforward. This complexity is reflective of the nuanced and varied nature of QCQP problems, where a one-size-fits-all theoretical analysis does not adequately capture the intricacies of different solver strategies and their outcomes.
>
> 2. **Common Practices in Current Literature:**
>
>     - In the literature of neighborhood search for mathematical programming, as are [2](https://openreview.net/forum?id=xEQhKANoVW&noteId=kBrRVMv-1k) [3](https://proceedings.mlr.press/v202/ye23e.html) [4](https://arxiv.org/abs/2107.10201)  [5](https://proceedings.neurips.cc/paper/2021/hash/fc9e62695def29ccdb9eb3fed5b4c8c8-Abstract.html) and [6](https://proceedings.mlr.press/v202/ye23e.html), empirical analysis (objective value ) rather than theoretical exploration is the predominant approach for evaluating methods' complexity and convergence.
>     - Such empirical analysis include:
> 	    - Comparison of objective value within fixed time
> 	    - Comparison of solution time with fixed objective value
> 	    - Improvement of objective value with respect to time
>     - This trend aligns with the practical challenges and the heuristic nature of such methods, where empirical results often provide more immediate and relatable insights into the algorithm’s performance under varied conditions.
>
> 3. **Our Methodology and Empirical Validation:**
>
>     - With our primary focus on large-scale QCQP problems, we have adopted a similar empirical approach () to validate our algorithm’s effectiveness. The convergence and performance of our solution framework have been thoroughly tested and demonstrated through extensive empirical analysis, as detailed in our submission (section 5 and appendix D.2). We have included comprehensive data and graphical representations to underscore the robustness and efficiency of our framework in handling large-scale problems. These results, we believe, not only attest to the practical utility of our approach but also provide a benchmark for future investigations.
>
> 4. **Future Theoretical Research:**
>
>     - We fully recognize the importance and value of a rigorous theoretical analysis in strengthening the scientific underpinnings of our methodology. The concerns raised in your review are indeed pertinent and align with one of our key future research directions. We are interested in exploring and developing a more comprehensive theoretical analysis of machine learning methods for mathematical programming.
>
> In summary, while our current submission emphasizes empirical results due to the nature and scope of the problems we are addressing, we are actively pursuing further theoretical insights into the field of neighborhood search methodologies.
>
> We are grateful for the opportunity to refine our work based on your feedback and look forward to any further guidance you may offer.

---

### Official Review · Reviewer_T75m · 2023-11-01

**Soundness:** 2 fair
**Presentation:** 1 poor
**Contribution:** 2 fair
**Rating:** 3
**Confidence:** 4

**Summary:**

The paper proposes a QCQP solver guided by neural networks. The QCQP instance is represented by a weighted hypergraph, which in turn is encoded by a weighted factor graph, which is an input to a graph neural network. This network predict solutions and guides a "neighborhood optimization", which focuses on a subset of variables and uses a convex relaxation of bilinear monomials (by McCormick) to get bounds. Several neighborhoods are searched in parallel. Sometimes some of their variables are exchanged, which is called a "cross-over". (Let me note that my understanding of the algorithm is not precise due to low clarity of the paper, see below).

In experimental evaluation, the method is compared to SOTA QCQP solvers (Gurobi and SCIP) on three problems: quadratic knapsack (QKP), quadratic independent set (QIS) and quadratic vertex cover (QVC) problems. While QKP is well-known, QIS and QVC were newly designed in the paper. The instances of the problems were generated randomly (to my understanding), no real instances (available online) are used. The experiments report that the new method is consistently faster than Gurobi and SCIP, which is more pronounced on larger instances.

**Strengths:**

The experimental results appear promising (but only on random QCQP instances).

**Weaknesses:**

I see two major weaknesses:

The text is unclear and hard to understand. The English is poor. The notation and mathematical formalism are clumsy and not well thought-over. The text is not well structured: sometimes a simple thing is given too much space but some substantial things are only in the supplement. There are a number of mistakes. Please see examples below.

In experimental evaluation, only random QCQP instances (generated by the authors) are used. No real instances available online (e.g.,  QAPLIB) are used. It is well-known that random instances of optimization problems tend to be easier than real one. Moreover, the QCQP problems that are used are described (in the supplement) in an insufficient and unclear way and appear to be artificial and sometimes meaningless. Let me give details:

QIS and QVP (described in sections A.4.1 and A.4.3) have not been proposed before, they were first proposed in the paper. However, their descriptions are confusing and insufficient and their formulations (7) and (10) not justified. No citations are given, neither to relevant similar problems nor their possible applications.

E.g., in A.4.1. the authors say that "the primary objective of [QIS] is to identify max. indep. set in the graph, which is a subset of vertices [...which..] do not belong to the same hyperedge". But this does not correspond to (7) (moreover, it is unclear whether in $\sum_{i,j\in e}$ in (8) one allows $i=j$). Moreover, this can be easily expressed by the linear constraints $\sum_{i\in e}x_i\le 1$. Morever, the authors introduced (randomly generated?) weights $a_i$ and $q_{ij}$ in the constraints without any explanation of their meaning. Last, it is not mentioned how the hypergraph $\cal H$ is generated when generating the instances.

In A.4.3., again the meaning of QVC formulation in (10) is not explained and it appears to make no sense.  What is the asterisk symbol in (10)? Why are the variables denoted $u,v$ rather than $i,j$ as before? How are the coefs $c_u,q_{u,v}$ chosen?

Quadratic Knapsack Problem is well-known but its definition in Section A.4.2 is different from the usual one (see, e.g., https://www.sciencedirect.com/science/article/pii/S0166218X06003878): in (8) there are multiple linear constraints on each individual binary variable $x_i$. This not justified and seems making no sense.

Due to these issues, I do not recommend to accept the paper. However, the approach can be potentially interesting. I recommend the authors to improve the text and experiments and perhaps consult the paper with someone skilled in writing good conference papers, and resubmit.

Some examples of mistakes / unclarities in the text:

- Section 2.1: The set $\cal X$ is mentioned only in problem formulation (1). In contrast, the set $\cal N$ does not appear in (1) (as as written below (1)). It is not clear if the QCQP has continuous or integer variables.

- Section 2.2: It is not described how the right-hand sides $b_i$ of MILP constraints are represented in the graph (despite the authors write that the MILP instance encoding by the graph is lossless). Similarly for the QCQP encoding in Section 4.1.1.

- Sec. 3.1: The definition of "bipartite hypergraph" is unclear. Note that in the literature, it can mean several different things.

- Sec. 3.1: Note that the result of "star expansion" is widely known in graphical models as "factor graph".

- Definition 1: The incidence matrix $\bf H$ and the "neighborhood relation" $N$ are almost the same thing and do not deserve a definition. I suggest using only one of these terms in the paper.

- Def. 2: One cannot define ${\cal N}_e(v)=\\{ e \mid vNe, \\; v\in V, \\; e\in E \\}$ because $v$ is given outside. That is, $v\in V$ must be omitted. Similarly in Def. 3.

- Def. 5: Shouldn't $v_i$ be $c_i$? Otherwise, the definitiions of ${\cal V}_x$ and ${\cal V}_c$ make no sense if the sets $\cal $N$ and $\cal M$ are not disjoint.

- Sec. A.4.1 (in the supplement): "we introduce the concept of hyperdges" - Why do you say this, after you used the concept of hypergraph many times before?

- Algorithm 1 in supplement: This should not be called an algorithm, it is just calculations of formulas. Moreover, these formulas were stated  already in the main text but with different notation. Note that Alg. 1 is not called in any of Algorithms 2,3,4,5.

- The QCQP formulation (1) is a minimization problem whether in the experiments (Tables 1,2) the problems are maximizations ones.

POST REBUTTAL: After rebuttal, the text was substantially revised by incorporating the reviewers' comments. I especially appreciate the new experiments on QPLIB and QAPLIB instances, on which the method appears to outperform off-the-shelf solvers. However, the formulation of some QCQP problems still makes little sense: for example, in QMKP (quadratic multiple knapsack problem) one knapsack can take only one object, whereas in the original formulation (Hiley & Julstrom, 2006) each knapsack can take more objects (up to its capacity). It is not described how the *graphs* (not coefficients) of the training instances are randomly generated. Last but not least, let me remark that the initial manuscript was in a very poor shape and I believe it is not a good practice to submit such papers and rely on reviewers to eliminate mistakes that could have been easily avoided with more effort. In summary, I am raising my evaluation by one level but I still do not find the paper not firm enough for acceptation.

**Questions:**

I have no concrete questions.

---

> ### Author Response · Authors · 2023-11-21
>
> Dear Reviewer,
>
> Thank you very much for your detailed comments and constructive criticisms. In response to your valuable feedback, we have meticulously reviewed the entire manuscript, making comprehensive revisions to enhance the clarity of the text, structure, and overall expression. We have also carefully refined and standardized the notation and mathematical definitions throughout the paper to ensure accuracy and coherence. Following these extensive revisions, we have resubmitted the paper. Additionally, we are supplementing our experiments with results from the QAPLIB dataset, which are now included in Appendix D. Our response is organized into three main sections: our motivation and experiment results, a thorough description of the problem formulations we have developed, and other detailed modifications and responses to your points.
>
>
>
> ### 1. Our Motivation for Experimental Evaluation and Results
>
> 1.1. **Motivation:**
>
>    - We opted for randomly generated problems, considering the current state of QCQP research and our primary objectives:
>      1. Machine learning methods have been used to optimize problems on a larger scale or at a faster speed. However, **the use of machine learning methods in solving QCQP is scarce, and existing datasets like QPLIB [1](https://link.springer.com/article/10.1007/s12532-018-0147-4) are limited.** QPLIB contains only 453 problems, each distinct and sourced from different distributions, unsuitable for machine learning applications. Meanwhile, other related learning methods in literature have also focused on **smaller scales** [2](http://arxiv.org/abs/2301.00306) or **quadratic problems (quadratic objective with linear constraints)** [3](https://link.springer.com/chapter/10.1007/978-3-319-93031-2_43) [4](https://pubsonline.informs.org/doi/abs/10.1287/ijoc.2022.1181?casa_token=Xb9xbDNG7BYAAAAA:aGPrsKwU_HVVq-Mu9Uv3iBoZD2Id4VNu_2ZfwlGENFg8TPz6YJbeI6W-i8Y1KIlkic0Z_0nqFL5F), which is unsuitable for benchmarking with our method.
>      2. **To validate our solution framework on different problems**, we introduced three problem forms: QKP, QIS, and QVC. Each form represents a different type of constraint and objective function, which we detail in the updated appendix A.4. The variety in test problems is aimed to provide a comprehensive evaluation of our algorithm's performance.
>      3. **To obtain large-scale problems for training and testing**, we used random methods to generate QCQP instances. This ensures problems of the same type and scale can be perceived as coming from the same distribution, making them suitable for neural network learning.
>      4. Despite the potentially lower complexity of randomly generated problems compared to real-world instances, the computation time taken by solvers like Gurobi and SCIP in our experiments attests to **the complexity of our generated problems**.
>
>    We thank you for suggesting the addition of QAP problems and have included them in our experiments in Appendix D.2.
>
> 1.2. **Experiment Results on more real-work problems:**
>
>    - Our experiments demonstrate that our method can address a variety of problems, showing its universality, and performs particularly well on large-scale problems.
>    - Tests on real-world problems, including QAPLIB and QPLIB, have been conducted. We trained neural network models on our generated problems and then tested them on real-world problems. Experiment settings and results are presented in Appendix D in the updated version, which further validates the effectiveness and efficiency of our framework.

---

> ### Author Response · Authors · 2023-11-21
>
> ### 2. Further Clarification on Problem Formulation
>
> Given the scarcity of the current benchmark dataset and problems, we generated random instances in the following. We have enhanced our descriptions of each problem form in Appendix A.4.
>
> **Quadratic Multiple Knapsack Problem (QMKP):**
>
> The QMKP problem [5](https://dl.acm.org/doi/abs/10.1145/1143997.1144096?casa_token=SYh1mdTyStsAAAAA:vImGiDvs4SgDbLysHgjth6_zKkURNwRKkhe1DHgItbyoZfTgl4c_G3qpJwwUT4BlJPjxueshUUCexwk) we study is a variant of the QKP problem. We initially used the abbreviation QKP, but following your suggestion, we have changed this to QMKP for clarity.
>
> The real-world significance of the QMKP problem involves optimizing the value of items with multiple attributes under certain constraints. An example would be maximizing the nutritional value of a day's diet, where some food combinations offer greater nutritional benefits (similar to a standard QKP), but each food item also has attributes like calories and sodium content that should not exceed certain limits.
>
> **QIS:**
>
> We developed the QIS (Quadratic Independent Set) problem by building on the foundation of the IS (Independent Set) problem, introducing quadratic non-convex constraints to give it practical significance. The classic IS problem is a typical combinatorial optimization challenge, assuming uniformity among all points and edges. However, in realistic network models, each node can have distinct attributes. To represent this, random weights $c_i$ are often introduced into the objective function to denote the attributes of node $i$.
>
> Considering that real-world network nodes might exhibit high-order relationships, we employed a hypergraph model where each hyperedge can include an arbitrary number of nodes. Additionally, these hyperedges may have non-linear relationships, so we introduced a function $f$ to represent these non-linear interactions. The coefficients $a_i$ and $q_{ij}$ are randomly generated from $U(0,1)$, and the inclusion of $|e|$ (the degree of the hyperedge) in our constraints is designed to ensure that these constraints are effective yet achievable, while also correlating them with the hyperedges.
>
> The QIS model has potential applications in scenarios involving hypergraph network models with nonlinear inter-node relationships. For instance, in wireless communication networks, particularly in frequency assignment, the challenge often lies in allocating frequencies to transmitters or channels in a manner that minimizes interference and maximizes network capacity or efficiency. This scenario can be conceptually similar to our QIS model, where it's essential to consider the (possibly nonlinear) interactions between various network elements. Other relevant applications include social network analysis and similar problems where complex relationships and attributes play a critical role.
>
>
> **QVC:**
>
> We derived the QVC problem from the standard vertex covering problem (appendix A.4.3), introducing quadratic terms in both the objective function and constraints. This represents one of the most challenging classes of QCQP problems and serves to evaluate the effectiveness of our solution framework. Based on our experiment results, QVC problems are the hardest problems among the three.
>
> **QAP:**
>
>    - We acknowledge your suggestion to use QAP problems as benchmarks. The reasons why we didn't test our method on QAP are as follows:
>      - QAP problems have been extensively studied [6](https://www.sciencedirect.com/science/article/pii/S0377221705008337?casa_token=p_j4KsAb5U8AAAAA:52eFVEmXW77ar44Ufa4oWTWrCKbMmaiZaG_nG8LELsqDYxOBfLNrJZyqn0mrM6kQNBZnBHkhkbzG). There are many methods [7](https://link.springer.com/article/10.1007/s12652-018-0917-x) targeting at QAP problems. Modeling the QAP as a QCQP problem and solving it using general solvers like Gurobi is much less efficient than using specific solution strategies.
>      - Our framework's ultimate goal is to predict optimal solutions and perform neighborhood search on **large-scale** and **general** QCQP problems, hence we did not include QAP problems in our dataset in the first place.
>    - Despite the inappropriateness, following your suggestion, we still included QAP in our benchmark problems to evaluate our method comprehensively. Results are given in Appendix D.2.

---

> ### Author Response · Authors · 2023-11-21
>
> ### 3. Mistakes/Uncertainties and Revisions
>
> 3.1. **Revisions Based on Your Suggestions:**
>
>  We have updated formulation (1) to more clearly reflect integer variables and included both maximization and minimization problems. We also added explanations for $b_i$ in sections 2.2 and 4.1.1 and revised the definition in Def 5. We have made language adjustments and streamlined the paper, integrating algorithm 1 into algorithm 2 for clearer relationships between algorithms. Further specific details of these improvements include:
>
> 3.2. **Clarification on Bipartite Hypergraph Definition:**
>
>    - We have refined the definition of bipartite hypergraph to avoid ambiguity, following standard definitions in literature ([8](https://epubs.siam.org/doi/abs/10.1137/1.9781611974331.ch126), [9](https://link.springer.com/article/10.1007/s10479-022-05073-9)).
>
> A hyper graph $\mathcal{H}$ is defined as
> $$
> \begin{aligned}
> &\mathcal{H} = (\mathcal{V}_1, \mathcal{V}_2, \mathcal{E})\text{ such that}\\
> &(1)\quad\mathcal{V}_1 \cap \mathcal{V}_2 = \emptyset,\\
> &(2)\quad |e\cap \mathcal{V}_1| = 1 \text{ and } e\cap \mathcal{V}_2 \neq \emptyset, \forall e \in \mathcal{E}.
> \end{aligned}
> $$
>
>
> 3.3. **On Definitions 1 and 2:**
>
>    - We initially used definitions consistent with the literature [10](https://ieeexplore.ieee.org/abstract/document/9795251?casa_token=0LL4IvxB06UAAAAA:WyiNXPIHsdqeZEXaD6PWtAMFUZteSAxlvUPj_5fOZrJ8jI_n0BkAtC6SpYM9ebhNuKJt_Fv7iDU) and [11](https://link.springer.com/10.1007/978-981-99-0185-2). To maintain clarity and consistency, we have retained the definitions of incidence matrix $H$ and inter-neighbor relation $N$. Following your advice, we have improved the definitions of $\mathcal{N}_e (v)$ and $\mathcal{N}_v (e)$ :
>
>      $$
>      \mathcal{N}_e(v) = \{ e | v\mathrm{N} e, e \in \mathcal{E} \}, \text{ for each } v \in \mathcal{V}.
>      $$
>
>      $$
>      \mathcal{N}_v(e) = \{ v | v\mathrm{N} e, v \in \mathcal{V} \}, \text{ for each }e \in \mathcal{E}.
>      $$
>
> We sincerely thank you for your valuable suggestions. We have uploaded the revised PDF version of our paper and hope for a fair reassessment of our work.
>
>
>
> **References**
> [1] Furini, Fabio, et al. "QPLIB: a library of quadratic programming instances." _Mathematical Programming Computation_ 11 (2019): 237-265.
> [2] Kannan, Rohit, Harsha Nagarajan, and Deepjyoti Deka. "Learning to accelerate the global optimization of quadratically-constrained quadratic programs." _arXiv preprint arXiv: 2301.00306_ (2022).
> [3] Bonami, Pierre, Andrea Lodi, and Giulia Zarpellon. "Learning a classification of mixed-integer quadratic programming problems." _Integration of Constraint Programming, Artificial Intelligence, and Operations Research: 15th International Conference, CPAIOR 2018, Delft, The Netherlands, June 26–29, 2018, Proceedings 15_. Springer International Publishing, 2018.
> [4] Bertsimas, Dimitris, and Bartolomeo Stellato. "Online mixed-integer optimization in milliseconds." _INFORMS Journal on Computing_ 34.4 (2022): 2229-2248.
> [5] Hiley, Amanda, and Bryant A. Julstrom. "The quadratic multiple knapsack problem and three heuristic approaches to it." _Proceedings of the 8th annual conference on Genetic and evolutionary computation_. 2006.
> [6] Loiola, Eliane Maria, et al. "A survey for the quadratic assignment problem." _European Journal of Operational Research_ 176.2 (2007): 657-690.
> [7] Abdel-Basset, Mohamed, et al. "A comprehensive review of quadratic assignment problem: variants, hybrids and applications." _Journal of Ambient Intelligence and Humanized Computing_ (2018): 1-24.
> [8] Annamalai, Chidambaram. "Finding perfect matchings in bipartite hypergraphs." _Proceedings of the twenty-seventh annual ACM-SIAM symposium on Discrete algorithms_. Society for Industrial and Applied Mathematics, 2016.
> [9] Aronshtam, Lior, Hagai Ilani, and Elad Shufan. "Perfect matching in bipartite hypergraphs subject to a demand graph." _Annals of Operations Research_ 321.1-2 (2023): 39-48.
> [10] Gao, Yue, et al. "HGNN+: General hypergraph neural networks." _IEEE Transactions on Pattern Analysis and Machine Intelligence_ 45.3 (2022): 3181-3199.
> [11] Dai, Qionghai, and Yue Gao. _Hypergraph Computation_. Springer Nature Singapore, 2023. [https://doi.org/10.1007/978-981-99-0185-2](https://doi.org/10.1007/978-981-99-0185-2).

---

### Author Response · Authors · 2023-11-23
**General response to all reviewers and the new revision**

Dear reviewers,

We sincerely appreciate the time and effort you have invested in reviewing our manuscript. Your insightful comments and constructive feedback have been invaluable in enhancing the quality and clarity of our work. We have carefully considered each point raised and have undertaken substantial revisions to address your concerns. Below is a summary of the key modifications and additions made to our manuscript:

1. **Revision of Dataset Size and Training Methodology:**
   - In response to your suggestion, we have critically reviewed the size of our dataset. We have explained our decision to maintain the current size, detailing the rationale behind our choice and the balance between dataset size, training time, and performance. Dataset details are added in Section 5 in the updated main paper.

2. **Enhanced Clarity and Precision in Writing:**
   - We have thoroughly revised the manuscript to improve clarity of expression, logical flow, and structural coherence. Special attention has been given to refining the notation and definitions related to mathematical concepts, ensuring accuracy and consistency throughout the paper.

3. **Updated Analysis on QPLIB and QAPLIB Experiments:**
   - We have expanded our experimental analysis, especially focusing on the QPLIB and QAPLIB datasets. New results and detailed discussions are now included in Appendix D, providing deeper insight into our method's performance and validate its applicability to real-world datasets.

4. **Incorporation of Area-Under-Curve Analysis and Running Time Comparisons:**
   - Thanks to your suggestion, we have plotted the area-under-curve divided by running time, presented in Appendix E.3, and conducted a comparison of objective values and running times against Gurobi and SCIP. These comparisons highlight the efficiency and effectiveness of our approach.

We hope that these revisions have significantly improved our manuscript and addressed the concerns you raised in your review. We remain open to any further guidance or clarifications and are eager to make any additional changes that might be necessary to further enhance our work.

Thank you once again for your valuable insights and guidance, which have been instrumental in refining our research.

---

### Meta-Review · Area_Chair_RNa1 · 2023-12-06

**Metareview:**

Summary: The paper presents a neural solver for quadratically constrained quadratic programs. A graph neural network predicts a solution, which is refined via neighbourhood optimization. Experimental results are shown on randomly generated instances of three types of problems.

+ ML for optimization is of interest to a wide audience.
+ The proposed method is novel.

- There are concerns regarding the experimental setup (see post-rebuttal comments of v1Pg and T75m)

**Justification For Why Not Higher Score:**

Despite the significant improvement in the clarity of the paper, the reviewers do not consider it to clean enough for acceptance in a top-tier ML conference.

There are also concerns regarding the experimental setup (see the post-rebuttal comments of reviewer T75m).

**Justification For Why Not Lower Score:**

N/A

---

### Decision · Program_Chairs · 2024-01-16

Reject